# Two forms of Opa1 cooperate to complete fusion of the mitochondrial inner-membrane

Yifan Ge[1], Xiaojun Shi[2†], Sivakumar Boopathy[1], Julie McDonald[1], Adam W Smith[2], Luke H Chao[1,3]*

[1]Department of Molecular Biology, Massachusetts General Hospital, Boston, United States; [2]Department of Chemistry, University of Akron, Akron, United States; [3]Department of Genetics, Harvard Medical School, Boston, United States

**Abstract** Mitochondrial membrane dynamics is a cellular rheostat that relates metabolic function and organelle morphology. Using an in vitro reconstitution system, we describe a mechanism for how mitochondrial inner-membrane fusion is regulated by the ratio of two forms of Opa1. We found that the long-form of Opa1 (l-Opa1) is sufficient for membrane docking, hemifusion and low levels of content release. However, stoichiometric levels of the processed, short form of Opa1 (s-Opa1) work together with l-Opa1 to mediate efficient and fast membrane pore opening. Additionally, we found that excess levels of s-Opa1 inhibit fusion activity, as seen under conditions of altered proteostasis. These observations describe a mechanism for gating membrane fusion.

*For correspondence:
chao@molbio.mgh.harvard.edu

Present address: †Rammelkamp Center for Research and Department of Medicine, MetroHealth System; Department of Physiology and Biophysics, School of Medicine, Case Western Reserve University, Ohio, United States

Competing interests: The authors declare that no competing interests exist.

## Introduction

Mitochondrial membrane fission and fusion is essential for generating a dynamic mitochondrial network and regenerative partitioning of damaged components via mitophagy (*Hoppins et al., 2007*). Membrane rearrangement is essential for organelle function (*Cipolat et al., 2006*; *Cogliati et al., 2013*) and contributes to diversity in mitochondrial membrane shape that can reflect metabolic and physiological specialization (*Nunnari and Suomalainen, 2012*; *Westermann, 2010*; *Anand et al., 2014*).

Mitochondrial membrane fusion in metazoans is catalyzed by the mitofusins (Mfn1/2) and Opa1 (the outer and inner membrane fusogens, respectively), which are members of the dynamin family of large GTPases (*Chen et al., 2003*; *Alexander et al., 2000*) (*Figure 1A*). An important series of in vitro studies with purified mitochondria showed that outer- and inner membrane fusion can be functionally decoupled (*Meeusen et al., 2006*; *Meeusen et al., 2004*). Outer membrane fusion requires Mfn1/2, while inner-membrane fusion requires Opa1. Loss of Opa1 function results in a fragmented mitochondrial network, loss of mitochondrial DNA, and loss of respiratory function (*MacVicar and Langer, 2016*; *Olichon et al., 2003*). Opa1 is the most commonly mutated gene in Dominant Optic Atrophy, a devastating pediatric condition resulting in degeneration of retinal ganglion cells. Mutations in Opa1 account for over a third of the identified cases of this form of childhood blindness (*Pesch et al., 2001*).

Like dynamin, Opa1 comprises a GTPase domain, helical bundle signaling element (BSE), and stalk region (with a membrane-interaction insertion) (*Figure 1B*) (*Schmid and Frolov, 2011*; *Ramachandran and Schmid, 2018*; *Faelber et al., 2019*). A recent crystal structure of the yeast orthologue of Opa1, Mgm1, revealed this membrane-interaction insertion is a 'paddle', which contains a series of hydrophobic residues that can dip into one leaflet of a membrane bilayer (*Faelber et al., 2011*).

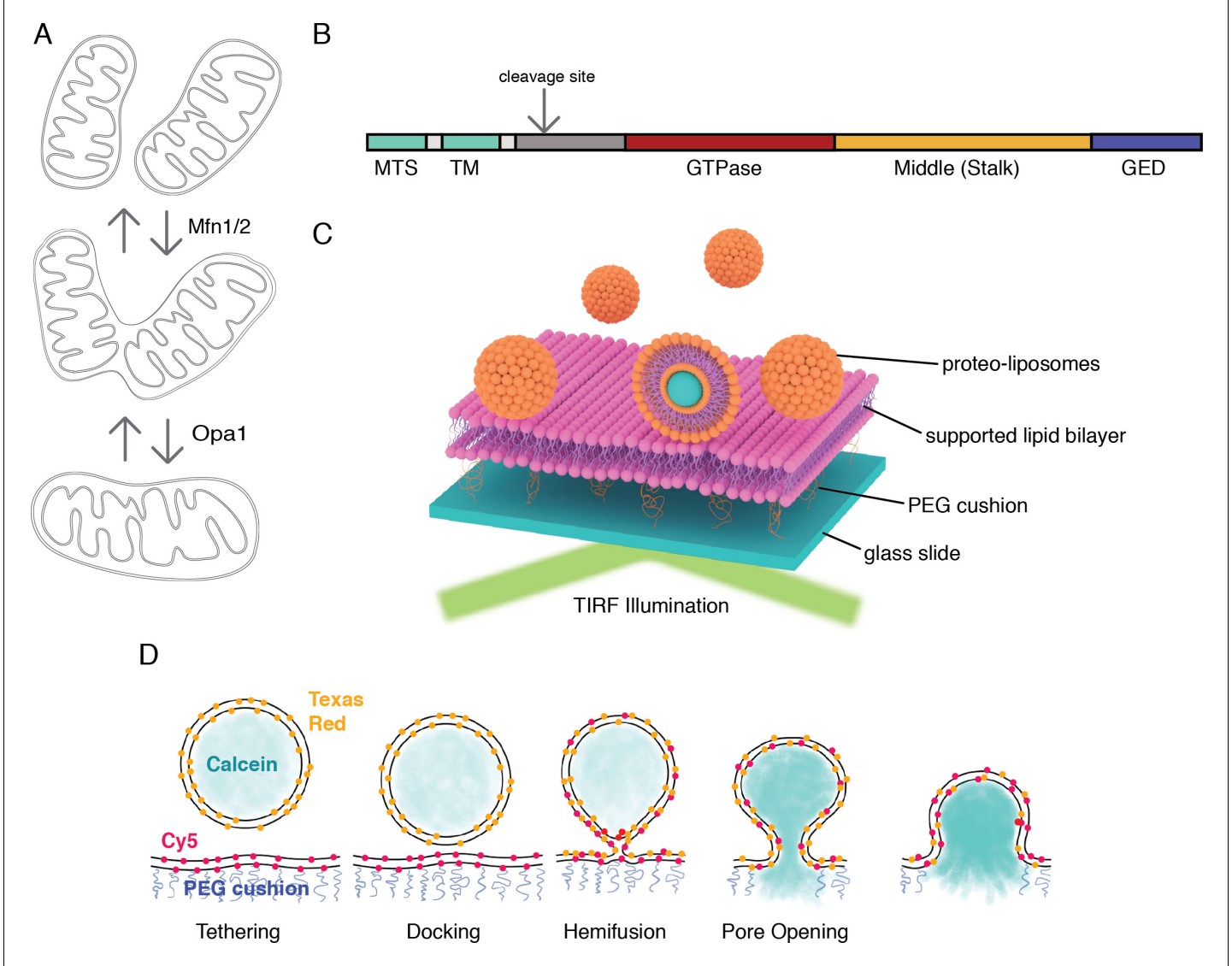

**Figure 1.** An in vitro assay for mitochondrial inner-membrane fusion. (A) Mitochondrial membrane fusion involves sequential outer and inner membrane fusion. The mitofusins (Mfn1/2) catalyze outer membrane fusion. In metazoans, mitochondrial inner-membrane fusion is mediated by Opa1. (B) Linear domain arrangement of l-Opa1. (C) Schema of the experimental setup. (D) Fusion assay. Membrane tethering, docking, lipid mixing, and content release can be distinguished using fluorescent reporters that specifically reflect each transition of the reaction.

Opa1 is unique for a dynamin family GTPase, because it is processed to generate two forms. The unprocessed, N-terminal transmembrane anchored, long form is called l-Opa1. The proteolytically processed short form, which lacks the transmembrane anchor, is called s-Opa1 (*Mishra et al., 2014*). Opa1 is processed by two proteases in a region N-terminal to the GTPase domain. Oma1 activity is stimulated by membrane depolarization (*Ehses et al., 2009*). Yme1L activity is coupled to respiratory state. Both forms of the protein (s-Opa1 and l-Opa1) can interact with cardiolipin, a negatively charged lipid enriched in the mitochondrial inner membrane. Opa1 GTPase activity is stimulated by association with cardiolipin (*Ban et al., 2010*).

Recent structural studies of Mgm1 focused on a short form, s-Mgm1 construct (*Faelber et al., 2019*). This analysis revealed a series of self-assembly interfaces in Mgm1's stalk region. One set of interactions mediates a crystallographic dimer, and a second set, observed in both the crystal and cryo-electron tomographic (cryo-ET) reconstructions, bridge dimers in helical arrays on membrane tubes with both positive and negative curvature. The s-Mgm1 membrane tubes that formed with negative curvature are especially intriguing, because of Opa1's recognized role in cristae regulation,

and the correspondence of the in vitro tube topology with cristae inner-membrane invaginations (*Meeusen et al., 2006*; *Frezza et al., 2006*). These self-assembled states were not mediated by GTPase-domain dimers.

Integrative biophysical and structural insights have revealed how dynamin nucleotide-state is coupled to GTPase-domain dimerization, stalk-mediated self-assembly and membrane rearrangement (*Faelber et al., 2011*; *Ford et al., 2011*; *Antonny et al., 2016*; *Chappie et al., 2010*). For Opa1, the opposite reaction (fusion) is also likely to result from nucleotide-dependent conformational changes, coupled domain rearrangement, and self-assembly necessary to overcome the kinetic barriers of membrane merger. Recent crystal structure and electron cryo-tomography reconstructions reveal self-assembly interfaces, and conformational changes that rearrange cristae membranes (*Faelber et al., 2019*). The specific fusogenic nucleotide hydrolysis-driven conformational changes remain to be distinguished.

Classic studies of Mgm1, the yeast orthologue of Opa1, show that both long and short forms are required for inner-membrane fusion (*DeVay et al., 2009*; *Herlan et al., 2003*). Studies by David Chan's group, using mammalian cells, also showed that both long and short forms of Opa1 are required (*Song et al., 2007*), and that knock-down of the Opa1 processing protease Yme1L results in a more fragmented mitochondrial network (*Mishra et al., 2014*). Since Yme1L activity is tied to respiratory state, supplying cells with substrates for oxidative phosphorylation shifts the mitochondrial network to a more tubular state. Importantly, Chan and colleagues cleanly demonstrate, with an in vitro purified mitochondria system using protease inhibitors and an engineered cleavage site that mitochondrial fusion is dependent on proteolytic processing (*Mishra et al., 2014*). In contrast, work from the Langer group showed l-Opa1 alone was sufficient for fusion when expressed in a YME1L -/-, OMA1 - /- background (*Anand et al., 2014*), indicating that Opa1 processing is dispensable for fusion. Over-expression of s-Opa1 in this background resulted in mitochondrial fragmentation, which was interpreted as a result of s-Opa1 mediated fission. Is proteolytic processing of Opa1 required for regulating fusion? Is s-Opa1 required for fusion?

In this study, we applied a TIRF-based supported bilayer/liposome assay (*Figure 1C*), to distinguish the sequential steps in membrane fusion that convert two apposed membranes into one continuous bilayer: tethering, membrane docking, lipid mixing (hemifusion) and content release (*Figure 1D*). This format allows control of protein levels for all components introduced into the system. Previous in vitro reconstitution studies from Ishihara and colleagues (*Ban et al., 2017*) were performed in bulk. The analysis we present here resolves individual fusion events in the TIRF field and is more sensitive than bulk measurements. In addition, our assay records kinetic data lost in ensemble averaging. Finally, the assay as applied here, can distinguish stages of fusion for individual liposomes. Tethering is observed when liposomes attach to the supported bilayer. Lipid mixing (hemifusion) is reported when a liposome dye (TexasRed) diffuses into the supported bilayer. Release of a soluble content dye (calcein) from within the liposome (loaded at quenched concentrations) indicates full pore opening. Our assay includes a content reporter dye in all conditions, so we can relate each intermediate with full fusion, for example, comparing instances where there may be lipid mixing, but no content release.

Using this in vitro reconstitution approach, we describe key mechanistic requirements for mitochondrial inner-membrane fusion. We report efficiency and kinetics for each step in Opa1-mediated fusion. These experiments describe the membrane activities of l-Opa1 alone, s-Opa1 alone, and l-Opa1:s-Opa1 together. We find that s-Opa1 and l-Opa1 are both required for efficient and fast pore opening, and present a mechanism for how the ratio of l-Opa1 and s-Opa1 levels regulate inner-membrane fusion. These results are compatible and expand the original yeast observations (*DeVay et al., 2009*), explain previous cellular studies (*Anand et al., 2014*; *Mishra et al., 2014*), and contextualizes recent in vitro observations (*Ban et al., 2017*). The data presented here unambiguously describe the activities of Opa1, contributing to a more complete model for how inner-membrane fusion is regulated.

## Results

### Assay validation

We purified long and short forms of human Opa1 expressed in *Pichia pastoris*. Briefly, Opa1 was extracted from membranes using n-dodecyl-β-D-maltopyranoside (DDM) and purified by Ni-NTA and Strep-tactin affinity chromatography, and size exclusion chromatography (*Figure 2A*). A series of short isoforms are observed in vivo (*MacVicar and Langer, 2016*; *Del Dotto et al., 2018*). In this study, we focused on a short form corresponding to the S1 isoform resulting from Oma1 cleavage (*Figure 2B*). GTPase activity of purified Opa1 was confirmed by monitoring free phosphate release (*Figure 2C and D*). Opa1 activity was enhanced by the presence of cardiolipin, consistent with previous reports (*Figure 2C and D*, *Figure 2—figure supplement 1*) (*Ban et al., 2010*).

We reconstituted l-Opa1 into 200 nm liposomes and supported bilayers generated by Langmuir-Blodgett/Langmuir-Schaefer methods (*Naumann et al., 2002*). l-Opa1 was added to liposomes and a supported bilayer at an estimated protein:lipid molar ratio of 1:5000 and 1:50000, respectively.

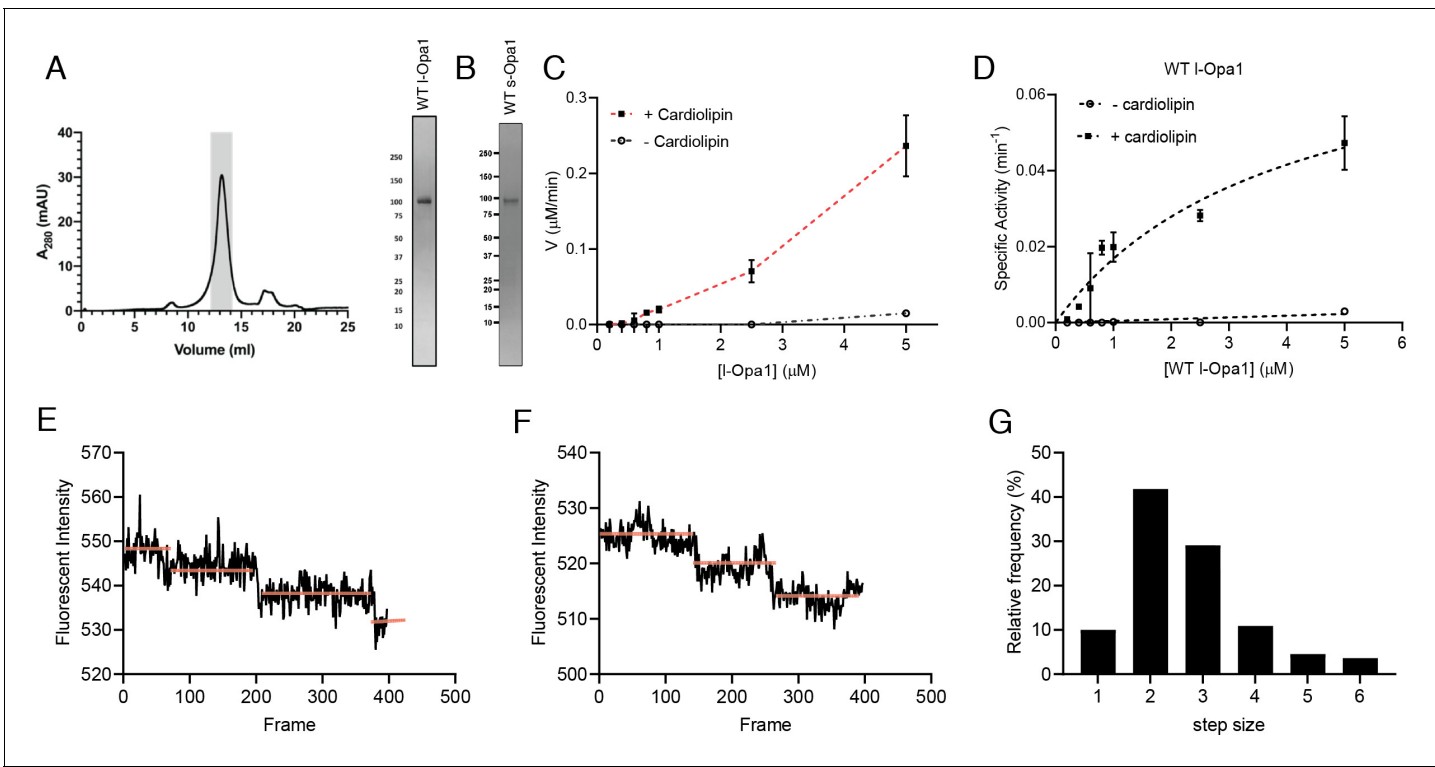

**Figure 2.** Reconstitution of l-Opa1. (**A**) Representative size-exclusion chromatograph and SDS-PAGE gel of human l-Opa1 purified from *P. pastoris*. (**B**) SDS-PAGE gel of human s-Opa1 purified from *P. pastoris*. l-Opa1 activity, with velocity (**C**) and specific activity (**D**) of GTP hydrolysis in the presence and absence of cardiolipin, while varying protein concentration of Opa1. Data are shown as mean ± SD, with error bars from three independent experiments. Representative single-liposome photobleaching steps (**E and F**) and histogram of step sizes (distribution for 110 liposomes shown) (**G**). The online version of this article includes the following source data and figure supplement(s) for figure 2:

**Source data 1.** Reconstitution of l-Opa1.
**Figure supplement 1.** GTP hydrolysis activity.
**Figure supplement 1—source data 1.** GTP hydrolysis activity.
**Figure supplement 2.** Liposome co-flotation.
**Figure supplement 2—source data 1.** Liposome co-flotation.
**Figure supplement 3.** Bilayer homogeneity and FCS.
**Figure supplement 3—source data 1.** Bilayer homogeneity and FCS.
**Figure supplement 4.** Blue native gels.
**Figure supplement 5.** FCS.
**Figure supplement 5—source data 1.** FCS.

Membranes comprised DOPE (20%), Cardiolipin (20%), PI (7%), and DOPC (52.8%). Reporter dyes (e.g. Cy5-PE, TexasRed-PE) were introduced into the supported bilayer and liposome membranes, respectively, at ~0.2% (mol). A surfactant mixture stabilized the protein sample during incorporation. Excess detergent was removed using Bio-Beads and dialysis. We confirmed successful reconstitution by testing the stability of l-Opa1 incorporation under high salt and sodium carbonate conditions, and contrasting these results with s-Opa1 peripheral membrane association (*Figure 2—figure supplement 2*).

We evaluated reconstitution of l-Opa1 into both the polymer-tethered supported lipid bilayers and proteoliposomes using two approaches. First, using Fluorescence Correlation Spectroscopy (FCS), we measured the diffusion of dye-conjugated lipids and antibody-labeled protein. FCS intensity measurements confirmed ~75% of l-Opa1 reconstituted into the bilayer in the accessible orientation. Bilayer lipid diffusion was measured as $1.46 \pm 0.12 \ \mu m^2/s$, while the diffusion coefficient of bilayer-reconstituted l-Opa1 was $0.88 \pm 0.10 \ \mu m^2/s$ (*Figure 2—figure supplement 3*), which is in agreement with previous reports of lipid and reconstituted transmembrane protein diffusion (*Siegel et al., 2011*). These measurements indicate the reconstituted l-Opa1 in the bilayer can freely diffuse, and has the potential to self-associate and oligomerize. Blue native polyacrylamide gel electrophoresis (BN-PAGE) analysis also show the potential for the purified material to self-assemble (*Figure 2—figure supplement 4*). FCS experiments were also performed on liposomes. FCS intensity measurements confirmed 86.7% of total introduced l-Opa1 successfully reconstituted into the liposomes. The diffusion coefficient of free antibody was $163.87 \pm 22.27 \ \mu m^2/s$. The diffusion coefficient for liposomes labeled with a lipid dye was $2.22 \pm 0.33 \ \mu m^2/s$, and the diffusion coefficient for l-Opa1 proteoliposomes bound to a TexasRed labeled anti-His antibody was $2.12 \pm 0.36 \ \mu m^2/s$ (*Figure 2—figure supplement 5*). Second, we measured the number of l-Opa1 incorporated into liposomes by fluorescent step-bleaching of single proteoliposomes (*Figure 2E and F*). We found an average step number of 2.7 for individual l-Opa1-containing proteoliposomes labeled with TexasRed conjugated anti-His antibody, when tethered to cardiolipin containing lipid bilayers (*Figure 2G*).

## Nucleotide-dependent bilayer tethering and docking

Using the supported bilayer/liposome assay sketched in *Figure 1C*, we found that l-Opa1 tethers liposomes in a homotypic fashion (*Figure 3A*), as reported by the appearance of TexasRed puncta in the TIRF field above a l-Opa1-containing bilayer. This interaction occurred in the absence of nucleotide (apo, nucleotide-free) but was enhanced by GTP. We next investigated requirements for Opa1 tethering. In the absence of cardiolipin, addition of GTP did not change the number of tethered particles under otherwise identical conditions (*Figure 3B*). In contrast, with cardiolipin-containing liposomes and bilayers, homotypic l-Opa1:l-Opa1 tethering is enhanced by GTP. Non-hydrolyzable analogues (GMPPCP) disrupt tethering (*Figure 3C*), and a hydrolysis-dead mutant (G300E) l-Opa1 shows little tethering (*Figure 3—figure supplement 1B*), supporting a role for the hydrolysis transition-state in tethering, as observed for atlastin (*Liu et al., 2015*; *O'Donnell et al., 2017*). Bulk light scattering measurements of liposome size distributions (by NTA Nanosight) show l-Opa1-mediated liposome clustering requires the presence of GTP (*Figure 3—figure supplement 2*). These bulk measurements show a GTP-dependent increase in observed particle size.

Ban, Ishihara and colleagues have observed a heterotypic, fusogenic interaction between l-Opa1 on one bilayer and cardiolipin in the opposing bilayer (*Ban et al., 2017*). Inspired by this work and our own observations, we considered if a heterotypic interaction between l-Opa1 and cardiolipin on the opposing membrane could contribute to l-Opa1-mediated tethering (*Figure 3D*). Indeed, we find that proteoliposomes containing l-Opa1 will tether to a cardiolipin-containing bilayer lacking any protein binding partner, presumably mediated by the lipid-binding 'paddle' insertion in the helical stalk region (*Faelber et al., 2019*). This tethering is cardiolipin-dependent, as l-Opa1 containing bilayers do not tether DOPC liposomes (*Figure 4—figure supplement 1B*).

We next measured whether s-Opa1, lacking the transmembrane anchor, could tether membranes via membrane binding interactions that bridge the two bilayers. We observe that s-Opa1 (added at a protein:lipid molar ratio of 1:5000) can tether cardiolipin liposomes to a cardiolipin-containing planar bilayer, as observed previously for Mgm1 (*Abutbul-Ionita et al., 2012*). Further, this s-Opa1 tethering is enhanced by the presence of GTP (*Figure 3E*). Previous reports observed membrane tubulation at higher concentrations of s-Opa1 (0.2 mg/ml, 1.67 nmol) (*Ban et al., 2010*). Under the lower s-Opa1 concentrations in our experiments (0.16 μg/ml, $2 \times 10^{-3}$ nmol), the supported bilayer

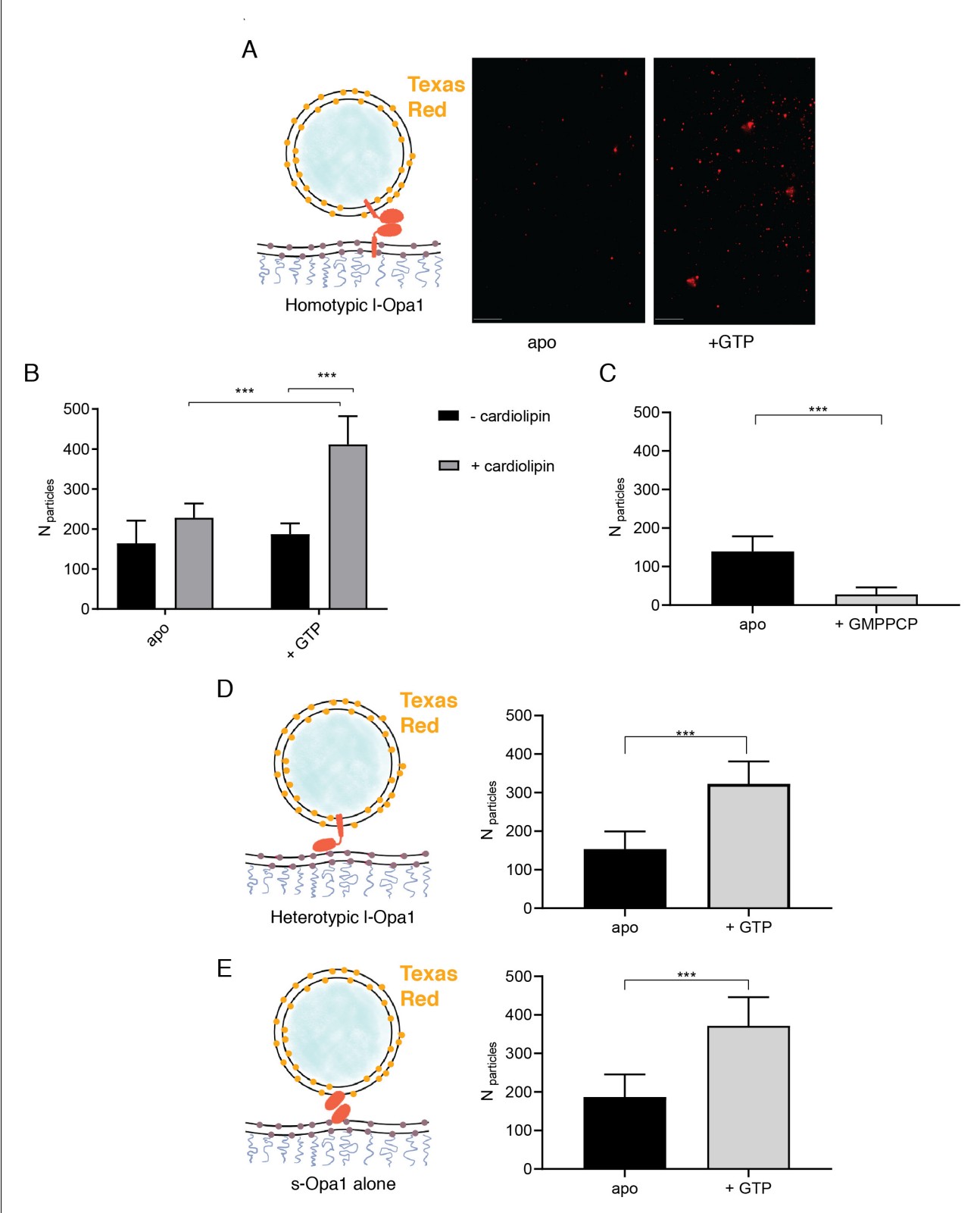

**Figure 3.** The number of liposomes tethered on the planar bilayers in a homotypic format (l-Opa1 on both bilayers) increases in the presence of GTP, when both bilayers contain cardiolipin. (**A**) Representative images of liposomes tethered on lipid bilayer (both containing cardiolipin) before (apo, or nucleotide free) and after GTP addition. Scale bar: 5 μm. (**B**) Bar graph: In the presence of cardiolipin, addition of GTP doubles the number of liposomes. (***p<0.001, two way ANOVA). (**C**) Addition of GMPPCP decreases amount of tethered l-Opa1 liposomes (apo, indicating nucleotide free)

*Figure 3 continued on next page*

*Figure 3 continued*

(p<0.005, two-way ANOVA). (**D**) l-Opa1 in the liposome bilayer alone is sufficient to tether liposomes to a cardiolipin containing bilayer. Tethering is enhanced in the presence of GTP (apo, indicating nucleotide free) (p<0.005, two-way ANOVA). (**E**) s-Opa1 tethers liposomes to a cardiolipin-containing bilayer. Number of tethered liposomes when both bilayer and liposomes contain 20% (mol) cardiolipin. Before addition of GTP (apo, indicating nucleotide-free), a moderate amount of liposome tethering was observed. The addition of GTP enhances this tethering effect (p<0.005, two-way ANOVA). Data are shown as mean ± SD. Error bars are from five independent experiments (>10 images across one bilayer per for each experiment).

The online version of this article includes the following source data and figure supplement(s) for figure 3:

**Source data 1.** Tethering.
**Figure supplement 1.** Effect of s-Opa1 competition on membrane tethering.
**Figure supplement 1—source data 1.** Effect of s-Opa1 competition on membrane tethering.
**Figure supplement 2.** Normalized relative and cumulative size distributions show cardiolipin containing proteoliposomes shift to larger sizes 1 hr following GTP addition (green trace), as measured by Nanosight light scattering.
**Figure supplement 2—source data 1.** Proteoliposome size distributions.

remains intact (before and after GTP addition), and we do not observe any evidence of tubular structures forming in the liposomes or bilayers.

Our experiments indicate that s-Opa1 alone can induce tethering. Is s-Opa1 competent for close docking of membranes? To answer this, we evaluated close bilayer approach using a reporter for when membranes are brought within FRET distances (~40–60 Å). This FRET signal reports on close membrane docking when a TexasRed conjugated PE is within FRET distance of a Cy5-conjugated PE. We observed a low FRET signal for tethered membranes, when the FRET pair is between two supported bilayers tethered via PEG spacer (average distance between the bilayer centers of ~7 nm) (*Minner et al., 2013*), compared to a single bilayer containing both of the FRET pair (*Figure 4— figure supplement 1A*). When l-Opa1 is present on both bilayers (homotypic arrangement), or on only one bilayer (heterotypic arrangement), efficient docking occurs in the presence of cardiolipin, as reported by a FRET signal with efficiencies of ~40% (*Figure 4B and C* and *Figure 4—figure supplement 1A*). Efficient homotypic docking requires GTP hydrolysis. GMPPCP prevents homotypic docking of l-Opa1, and abolishes the heterotypic l-Opa1 signal) (*Figure 4A*). In contrast, s-Opa1 alone does not bring the two bilayers within FRET distance, consistent with observations for s-Mgm1 tethered bilayers (*Figure 4A*) (*Abutbul-Ionita et al., 2012*). The distances between two paddles in the s-Mgm1 dimer is ~120 Å. Tethering mediated by two paddle interactions would be compatible with our observed low FRET signal when s-Opa1 engages two bilayers (*Faelber et al., 2011*).

## Hemifusion and pore opening

We find that l-Opa1, when present on only one bilayer, in a heterotypic format, can mediate close membrane docking (*Figure 4A*). To quantify hemifusion (lipid exchange), we measured the fluorescence intensity decay times for the liposome dye (TexasRed) as it diffuses into the bilayer during lipid mixing. Analysis of particle dye diffusion kinetics shows that in this heterotypic format, l-Opa1 can induce hemifusion (*Figure 5A*). The hemifusion efficiency (percentage of total particles where the proteoliposome dye diffuses into the supported bilayer) for heterotypic l-Opa1 was <5% (*Figure 6A*). Previously published in vitro bulk liposome-based observations for heterotypic l-Opa1 lipid mixing observed hemifusion efficiencies of 5–10%, despite higher protein copy number per liposome (*Ban et al., 2017*). We next compared homotypic l-Opa1 catalyzed hemifusion and observed shorter mean dwell times than heterotypic l-Opa1 (*Figure 5B and C*, *Figure 5—figure supplement 1*). In our assay, we observe homotypic l-Opa1 induces hemifusion more efficiently than heterotypic l-Opa1. We measured a homotypic l-Opa1 hemifusion efficiency of ~15% (*Figure 6A*). For comparison, in vitro measurements of viral membrane hemifusion, show efficiencies of ~25–80% (*Chao et al., 2014*; *Ivanovic et al., 2013*). This comparison is imperfect, as viral particles have many more copies of their fusion proteins on their membrane surface and viral fusogens do not undergo multiple cycles of nucleotide hydrolysis, like Opa1.

Following hemifusion, pore opening is the key step where both leaflets merge and content from the two compartments can mix. We observed pore opening by monitoring content dye (calcein) release under these conditions (*Rawle et al., 2011*). Of all homotypic tethered particles,~18% undergo hemifusion. Of these particles undergoing hemifusion, approximately half proceed to full fusion (8% of all homotypic tethered particles). Both s-Opa1 alone (at 0.16 µg/ml, or $2 \times 10^{-3}$ nmol

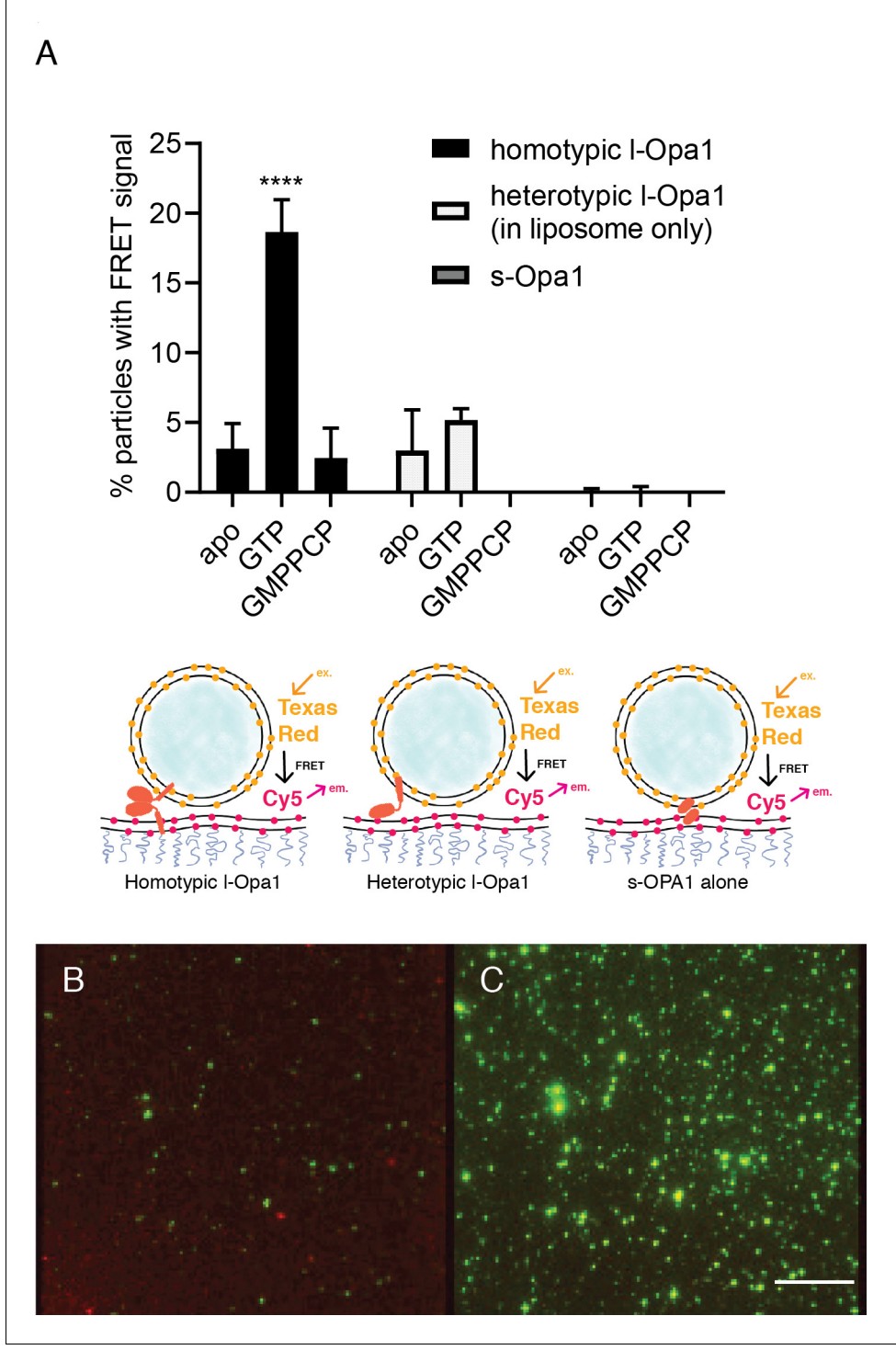

**Figure 4.** Docking. (**A**) Homotypic l-Opa1 docks liposomes in a GTP-hydrolysis dependent manner. s-Opa1, alone is insufficient to closely dock liposomes. l-Opa1 in a heterotypic format (on the liposome alone) is competent to closely dock to a bilayer, but this docking is not stimulated by nucleotide. Bar graphs shown as mean ± SD (p<0.0001, one-way ANOVA). Error bars are from 3 to 5 independent experiments (each experiment with >150 particles in a given bilayer). (**B**) In the presence of cardiolipin on both bilayers, FRET signal reports on close liposome docking mediated by l-Opa1. Left: Green = Cy5 emission signal upon excitation at 543 (TexasRed excitation). Red = Cy5 emission signal in membrane upon excitation at 633 (Cy5 excitation). Right: Green = TexasRed emission upon excitation at 543 nm (TexasRed excitation). Scale bar: 5 μm.

*Figure 4 continued on next page*

*Figure 4 continued*
The online version of this article includes the following source data and figure supplement(s) for figure 4:

**Source data 1.** Docking.
**Figure supplement 1.** Docking.
**Figure supplement 1—source data 1.** Docking.

concentration), or l-Opa1 in the heterotypic format did not release content (*Figure 6A*). In contrast, ~8% of homotypic l-Opa1:l-Opa1 particles undergo pore opening and content release. These observations indicate, l-Opa1 alone is sufficient for pore opening, while s-Opa1 alone or heterotypic l-Opa1 are insufficient for full fusion.

## Ratio of s-Opa1:l-Opa1 regulate pore opening efficiency and kinetics

Our observation that l-Opa1 is sufficient for pore opening leaves open the role of s-Opa1 for fusion. Previous studies suggest an active role for s-Mgm1 (the yeast orthologue of s-Opa1) in fusion (*DeVay et al., 2009*). In this work, l-Mgm1 GTPase activity was dispensable for fusion in the presence of wild-type s-Mgm1 (*DeVay et al., 2009*). Work in mammalian cells suggests different roles for s-Opa1. Studies from the Chan group showed Opa1 processing helps promote a tubular mitochondrial network (*Mishra et al., 2014*). In contrast, other studies showed upregulated Opa1 processing in damaged or unhealthy mitochondria, resulting in accumulation of s-Opa1 and fragmented mitochondria (*Mishra et al., 2014*; *Ban et al., 2017*; *Griparic et al., 2007*). The interpretation of the latter experiments was that, in contrast to the yeast observations, s-Opa1 suppresses fusion activity. Furthermore, studies using Opa1 mutations that abolish processing of l-Opa1 to s-Opa1 suggest the short form is dispensable for fusion, and s-Opa1 may even mediate fission (*Lee et al., 2017*; *Baker et al., 2014*). These different, and at times opposing, interpretations of experimental observations have been difficult to reconcile.

To address how s-Opa1 and l-Opa1 cooperate, we added s-Opa1 to the l-Opa1 homotypic supported bilayer/liposome fusion experiment. l-Opa1-only homotypic fusion has an average dwell time of 20 s and an efficiency of ~10% (*Figure 6B–E* and *Figure 6—figure supplement 1*). Upon addition of s-Opa1, we observe a marked increase in pore opening efficiency, reaching 80% at equimolar l-Opa1 and s-Opa1 (*Figure 6B*). At equimolar levels of s-Opa1, we also observe a marked change in pore opening kinetics, with a four-fold decrease in mean dwell time (*Figure 6C*). The efficiency peaks at an equimolar ratio of s-Opa1 to l-Opa1, showing that s-Opa1 cooperates with l-Opa1 to catalyze fast and efficient fusion. When s-Opa1 levels exceed l-Opa1 (at a 2:1 ratio or greater), particles begin to detach, effectively reducing fusion efficiency. This is consistent with a dominant negative effect, where s-Opa1 likely disrupts the homotypic l-Opa1:l-Opa1 interaction. We quantified particle untethering, and observe a dose-dependent detachment of l-Opa1:l-Opa1 tethered particles with the addition of G300E s-Opa1 (*Figure 3—figure supplement 1A*).

## Discussion

Our experiments suggest different assembled forms of Opa1 represent functional intermediates along the membrane fusion reaction coordinate, each of which can be a checkpoint for membrane fusion and remodeling. We show that s-Opa1 alone is sufficient to mediate membrane tethering but is unable to dock and merge lipids in the two bilayers, and thus, insufficient for hemifusion (*Figure 7A*). In contrast, l-Opa1, in a heterotypic format, can tether and hemifuse bilayers, but is unable to transition through the final step of pore opening (*Figure 7B*). Homotypic l-Opa1 can hemifuse membranes and mediate low levels of pore opening (*Figure 7C* i.). However, our results show that s-Opa1 and l-Opa1 together, synergistically catalyze efficient and fast membrane pore opening (*Figure 7C* ii.). Importantly, we find that excess levels of s-Opa1 are inhibitory for pore opening, providing a means to down-regulate fusion activity (*Figure 7C* iii.).

Our model proposes that l-Opa1:s-Opa1 stoichiometry gates the final step of fusion, pore opening. Electron tomography studies of mitofusin show a unevenly distributed ring of proteins clustering at an extensive site of close membrane docking, but only local regions of pore formation (*Brandt et al., 2016*). Our study is consistent with local regions of contact and low protein copy

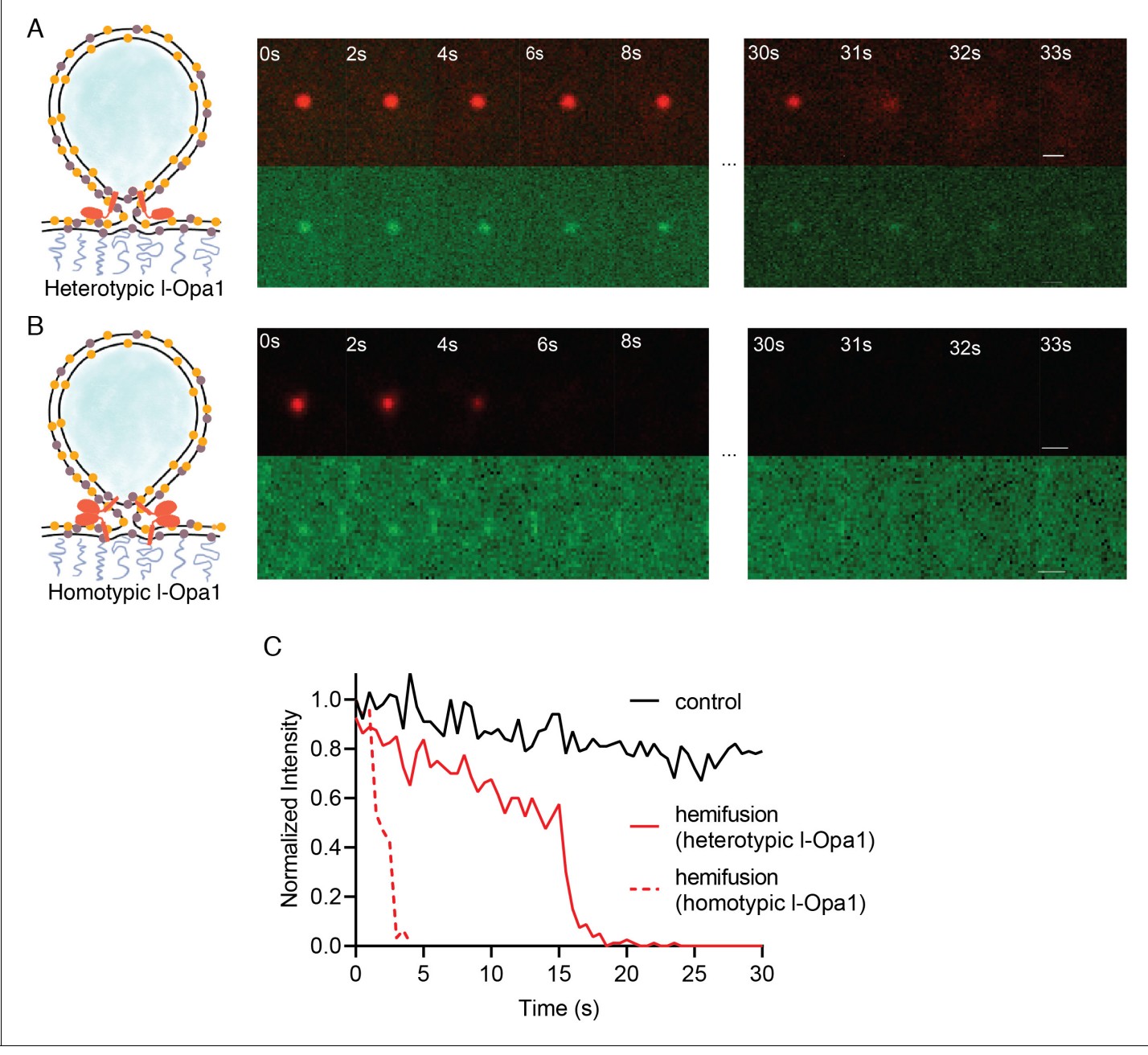

**Figure 5.** Hemifusion. (**A**) Heterotypic hemifusion. Top panels: time trace of proteo-liposome lipid dye diffusion (TexasRed). Bottom panels: no content release is observed for this particle (calcein signal remains quenched). Scale bar: 1 μm. (**B**) Homotypic hemifusion. Top panels: time trace of liposome lipid dye diffusion (TexasRed). Bottom panels: no content release is observed for this particle (calcein signal remains quenched). Scale bar: 1 μm. (**C**) Representative intensity traces of a control particle not undergoing fusion (black), with heterotypic hemifusion event (solid red), and homotypic hemifusion event (dotted red).

The online version of this article includes the following source data and figure supplement(s) for figure 5:

**Source data 1.** Heterotypic and homotypic hemifusion.

**Figure supplement 1.** Additional kinetic traces for hemifusion curves under homotypic (**A**) and heterotypic (**B**) Opa1 hemifusion conditions.

**Figure supplement 1—source data 1.** Additional hemifusion kinetic traces.

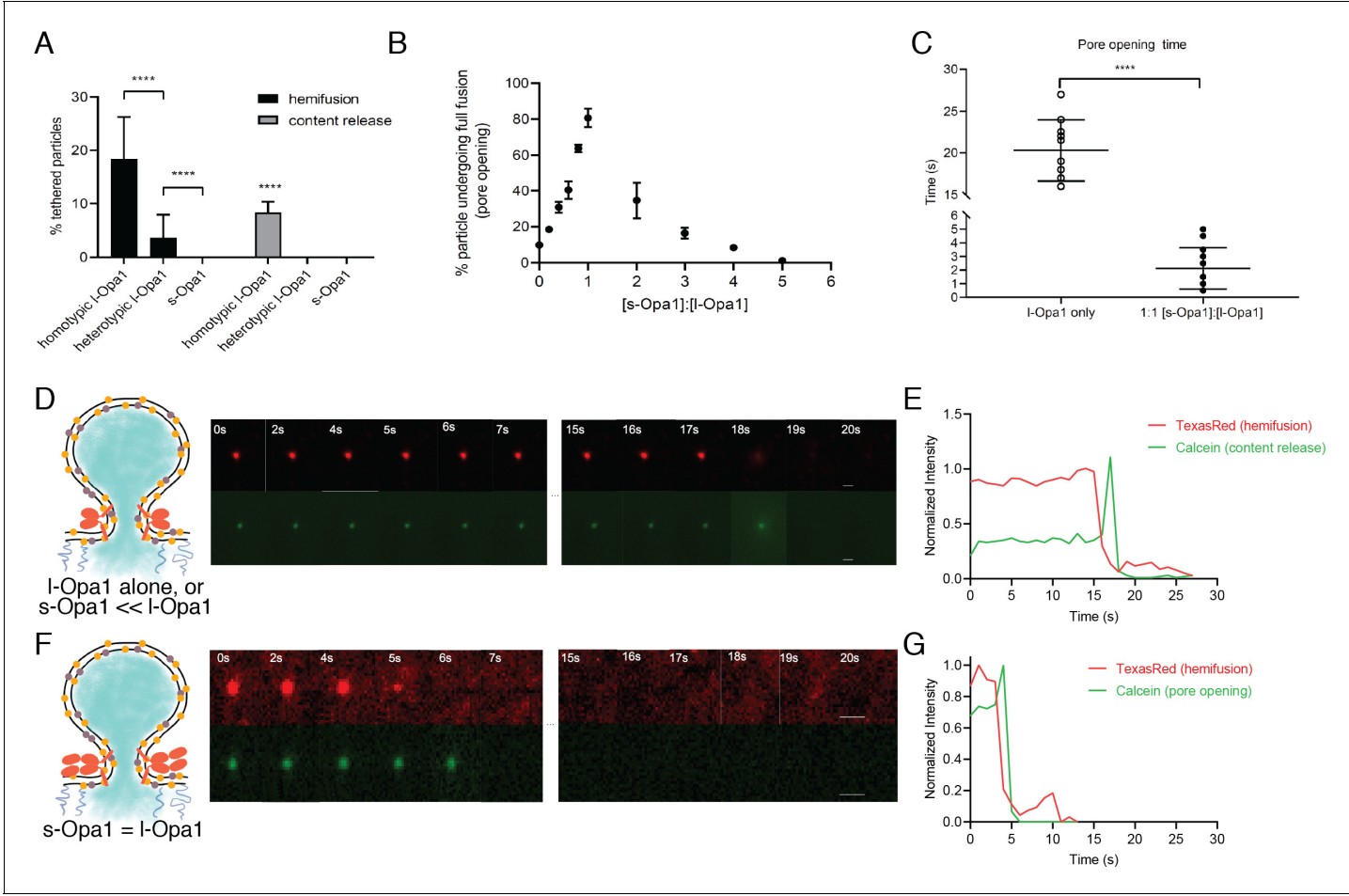

**Figure 6.** Hemifusion and full fusion. (**A**) Hemifusion (lipid mixing) and full fusion (content release and pore opening) efficiency for homotypic l-Opa1, heterotypic l-Opa1 and s-Opa1 (p<0.001, two-way ANOVA). Bar graphs shown as mean ± SD. Error bars are from five different experiments (50–200 particles were analyzed per bilayer in each experiment). B. Full fusion (pore opening) efficiency at different s-Opa1:l-Opa1 ratios. Data are shown as mean ± SD. Error bars are from 4 to 6 experiments (80–150 particles per bilayer in each experiment). The significance of the data was confirmed using one-way ANOVA (Prism 8.3) where p<0.0001. C. Mean pore opening time in the absence of s-Opa1 and at equimolar s-Opa1. Significance of the difference was confirmed using t-test (Prism 8.3, p<0.0001). D. Representative hemifusion and pore opening fluorescence time series for homotypic l-Opa1 experiment, in the absence of s-Opa1, top and bottom panels, respectively. Scale bar: 1 μm. E: representative traces of TexasRed (liposome signal) and calcein (content signal) intensity for homotypic l-Opa1 experiment. F. Representative hemifusion and pore opening fluorescence traces for a homotypic l-Opa1 experiment in the presence of equimolar s-Opa1. Scale bar: 1 μm. G: Representative trace of TexasRed (liposome signal) and calcein (content signal) intensity for a homotypic l-Opa1 experiment in the presence of equimolar s-Opa1.

The online version of this article includes the following source data and figure supplement(s) for figure 6:

**Source data 1.** Hemifusion and pore opening.

**Figure supplement 1.** Additional kinetic traces for hemifusion and pore opening under homotypic l-Opa1 conditions (A), homotypic l-Opa1, and l-Opa1 + s-Opa1 (1:1) (B) conditions.

**Figure supplement 1—source data 1.** Additional kinetic traces.

number mediating lipid mixing and pore formation (*Zick et al., 2009*). Our study would predict that s-Opa1 enrichment in regions of the mitochondrial inner-membrane would suppress fusion. This study did not explore the roles of s-Opa1 assemblies (helical or 2-dimensional) in fusion (*Faelber et al., 2019*). Cellular validation of our proposed model, and other states, will require correlating l-Opa1:s-Opa1 ratio and protein spatial distribution with fusion efficiency and kinetics. This study focused on isoform 1 of l-Opa1 and the S1 form of s-Opa1. The behavior of other Opa1 splice isoforms, which vary in the processing region, remains another important area for future investigation (*Wai et al., 2016*).

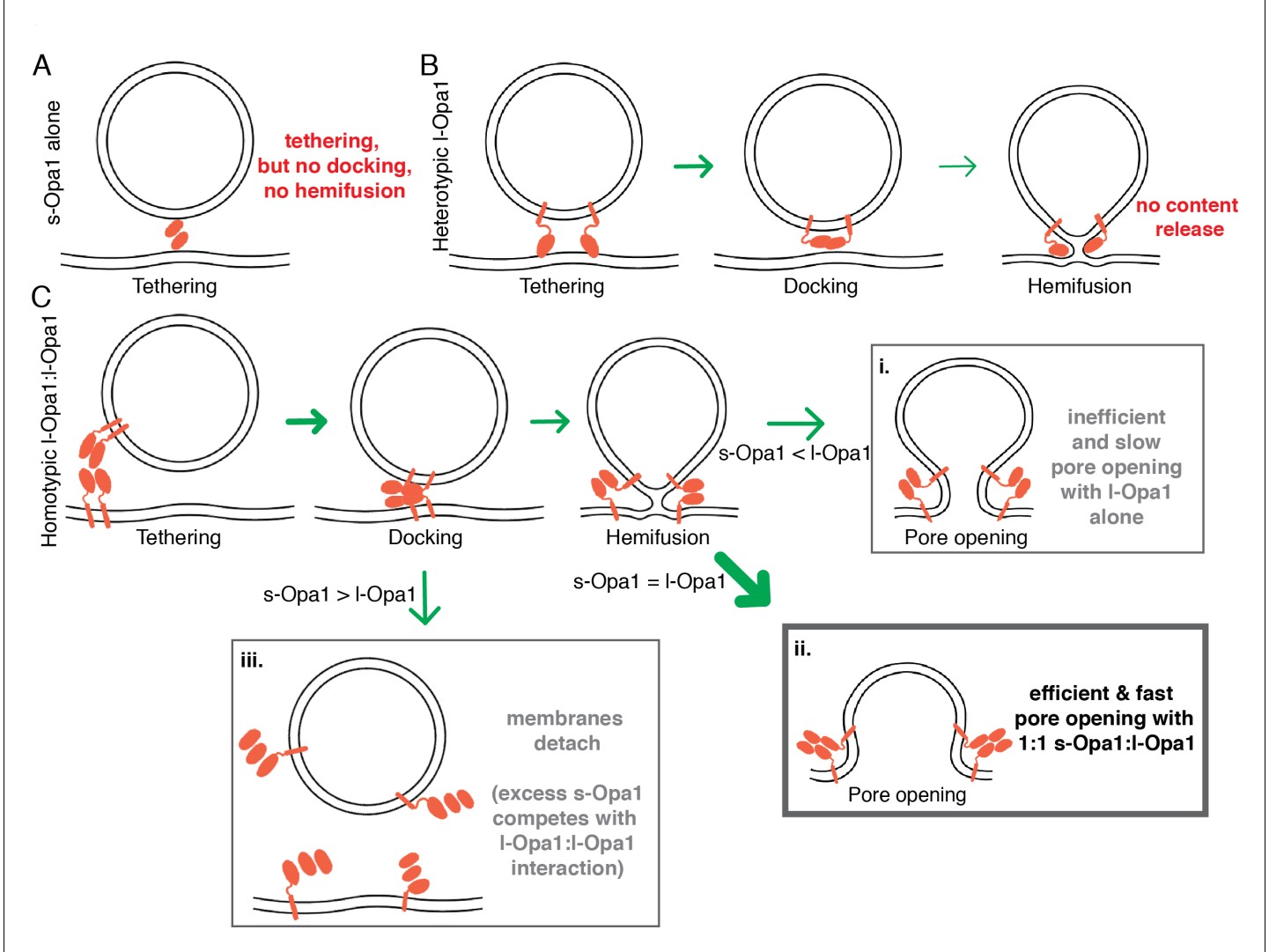

**Figure 7.** Summary model for modes of regulation in Opal-mediated membrane fusion. (**A**) s-Opa1 alone is capable of tethering bilayers, but insufficient for close membrane docking and hemifusion. (**B**) l-Opa1, in a heterotypic arrangement, can tether bilayers, and upon GTP stimulation promote low levels of lipid mixing, but no full fusion, pore opening or content release. (**C**) Homotypic l-Opa1-l-Opa1 tethered bilayers can mediate full content release (i). This activity is greatly stimulated by the presence of s-Opa1, with peak activity at 1:1 s-Opa1:l-Opa1 (ii). Excess levels of s-Opa1 suppress fusion, likely through competing with the l-Opa1-l-Opa1 homotypic tethering interface (iii).

The results and model presented here help resolve the apparent contradicting nature of the Chan and Langer cellular observations. As observed by the Langer group, l-Opa1 alone in our system, is indeed sufficient for full fusion, albeit at very low levels (*Anand et al., 2014*). The activity of unprocessed Opa1 was not ruled out in previous studies of Chan and colleagues (*Mishra et al., 2014*). In contrast to the Langer group's conclusions, we find that s-Opa1 strongly stimulates l-Opa1-dependent fusion activity, independent of the Yme1L processing reaction (*Mishra et al., 2014*). Under conditions of s-Opa1 overexpression, Langer et al. observed, a fragmented mitochondrial network. We do not see any evidence for fission or fusion, for s-Opa1 alone, under our reconstitution conditions. Instead, our data and model suggest this is due to s-Opa1 disrupting l-Opa1 activity, swinging the membrane dynamics equilibrium toward fission.

Mitochondrial dysfunction is often associated with Opa1 processing (*Duvezin-Caubet et al., 2006*). The activity of the mitochondrial inner-membrane proteases, Yme1L and Oma1, is regulated by mitochondrial matrix state, thereby coupling organelle health to fusion activity (*Anand et al., 2014*; *Baker et al., 2014*; *Duvezin-Caubet et al., 2006*; *Rainbolt et al., 2016*; *Ishihara et al.,*

*2006*). Basal levels of Opa1 cleavage are observed in healthy cells (*Mishra et al., 2014*). Upon respiratory chain collapse and membrane depolarization increased protease activity elevates s-Opa1 levels, downregulating fusion (*Baker et al., 2011*). Our results point to the importance of basal Opa1 processing, and are consistent with observations that both over-processing and under-processing of l-Opa1 can result in a loss of function (*Anand et al., 2014*).

Key questions remain in understanding the function of different Opa1 conformational states, and the nature of a fusogenic Opa1 complex. Recent structural studies show s-Mgm1 self-assembles via interfaces in the stalk region (*Faelber et al., 2019*; *Zhang et al., 2019*). The nucleotide-independent tethering we observe also implicate stalk region interactions, outside of a GTPase-domain dimer, in membrane tethering. How does nucleotide hydrolysis influence these interactions during fusion? Outstanding questions also remain in understanding the cooperative interplay between local membrane environment, s-Opa1, and l-Opa1. Could the cooperative activity of l-Opa1 and s-Opa1 be mediated by direct protein-protein interactions, local membrane change, or both? Could tethered states (e.g. homotypic l-Opa1 or heterotypic l-Opa1) bridge bilayers to support membrane spacings seen in cristae? Answers to these questions, and others, await further mechanistic dissection to relate protein conformational state, in situ architecture and physiological regulation.

# Materials and methods

**Key resources table**

| Reagent type (species) or resource | Designation | Source or reference | Identifiers | Additional information |
|---|---|---|---|---|
| Chemical compound, drug | 18:1 (Δ9-Cis) PC (DOPC) | Avanti Polar lipids | Cat #: 850375P | |
| Chemical compound, drug | 1′,3′-bis[1,2-dioleoyl-sn-glycero-3-phospho]-glycerol (sodium salt) | Avanti Polar lipids | Cat #: 710335P | |
| Chemical compound, drug | 1,2-dioleoyl-sn-glycero-3-phosphoethanolamine-N-[methoxy(polyethylene glycol)—2000] (ammonium salt) | Avanti Polar lipids | Cat #: 880130P | |
| Chemical compound, drug | L-α-lysophosphatidylinositol (Liver, Bovine) (sodium salt) | Avanti Polar lipids | Cat #: 850091P | |
| Chemical compound, drug | 1-palmitoyl-2-oleoyl-sn-glycero-3-phosphoethanolamine | Avanti Polar lipids | Cat #: 850757P | |
| Chemical compound, drug | Texas Red 1,2-Dihexadecanoyl-sn-Glycero-3-Phosphoethanolamine, Triethylammonium Salt (Texas Red DHPE) | ThermoFisher Scientific | Cat #: T1395MP | |
| Chemical compound, drug | 1,2-dioleoyl-sn-glycero-3-phosphoethanolamine-N-(Cyanine 5) | Avanti polar lipid | Cat #: 810335C1mg | |
| Chemical compound, drug | Calcein | Sigma-Aldrich | Cat #: C0875; PubChem Substance ID: 24892279 | |
| Strain | *Pichia pastoris* SMD1163 (*his4,pep, prb1*) | Rapoport lab; Harvard Medical School. | | |
| Recombinant DNA reagent | pPICZ A-TwinStrep-lOPA1-$H_{10}$ | GenScript | | plasmid to transform and express human WT l-Opa1 (isoform1). |
| Recombinant DNA reagent | pPICZ A-TwinStrep-sOPA1-$H_{10}$ | GenScript | | plasmid to transform and express human WT s-Opa1 (s1). |

*Continued on next page*

*Continued*

| Reagent type (species) or resource | Designation | Source or reference | Identifiers | Additional information |
|---|---|---|---|---|
| Recombinant DNA reagent | pPICZ A-TwinStrep-lOPA1 (G300E)-H$_{10}$ | GenScript | | plasmid to transform and express G300E mutant of l-Opa1 (isoform 1). |
| Recombinant DNA reagent | pPICZ A-TwinStrep-sOPA1 (G300E)-H$_{10}$ | GenScript | | plasmid to transform and express G300E mutant of s-Opa1 (s1). |
| Antibody | Rabbit Anti-Opa1 antibody | NOVUS BIOLOGICALS | Cat #: NBP2-59770 | Western Blot 2 ug/ml |
| Antibody | Mouse 6x-His Tag Monoclonal Antibody (HIS.H8) | ThermoFisher Scientific | Cat #: MA1-21315 | Western Blot 1:2000 |
| Antibody | Mouse StrepMAB-Classic, HRP conjugate (2-1509-001) | IBA Lifesciences | Cat #: 2-1509-001 | Western Blot 1:2500/1:32000 |
| Antibody | Rabbit IgG HRP Linked Whole Ab | SIGMA-ALDRICH INC | Cat #: GENA934-1ML | |
| Antibody | Mouse IgG HRP Linked Whole Ab | SIGMA-ALDRICH INC | Cat #: GENA931-1ML | |
| Chemical compound, drug | GTP Disodium salt | SIGMA-ALDRICH INC | Cat #: 10106399001 | |
| Commercial assay, kit | EnzChek Phosphate Assay Kit | ThermoFisher Scientific | Cat #: E6646 | |
| Chemical compound, drug | GppCp (Gmppcp), Guanosine-5'-[(β,γ)-methyleno]triphosphate, Sodium salt | Jena Bioscience | Cat #: NU-402–5 | |
| Chemical compound, drug | n-Dodecyl-β-D-Maltopyranoside | Anatrace | Cat #: D310 25 GM | |
| Chemical compound, drug | n-Octyl-α-D-Glucopyranoside | Anatrace | Cat #: O311HA 25 GM | |
| Chemical compound, drug | Lauryl Maltose Neopentyl Glycol (LMNG) | Anatrace | Cat #: NG310 | |
| Chemical compound, drug | LMNG-CHS Pre-made solution | Anatrace | Cat #: NG310-CH210 | |
| Chemical compound, drug | Zeocin | Invivogen | Cat #: ant-zn-1p | |
| Chemical compound, drug | Ni-NTA | Biorad | Cat #: 7800812 | |
| Chemical compound, drug | StrepTactin XT | IBA Lifesciences | Cat #: 2-4026-001 | |
| Chemical compound, drug | Biotin | IBA Lifesciences | Cat #: 2-1016-005 | |
| Chemical compound, drug | Superose 6 Increase 10/300 GL | GE | Cat #: 29091596 | |
| Chemical compound, drug | TEV protease | Prepared in lab, purchased from GenScript | Cat #: Z03030 | |
| Chemical compound, drug | Benzonase Nuclease | Sigma-Aldrich | Cat #: E1014 | |
| Chemical compound, drug | Protease inhibitor cocktail | Roche | Cat #: 05056489001 | |
| Chemical compound, drug | Leupeptin | Sigma-Aldrich | Cat #: L2884 | |

*Continued on next page*

*Continued*

| Reagent type (species) or resource | Designation | Source or reference | Identifiers | Additional information |
|---|---|---|---|---|
| Chemical compound, drug | Pepstatin A | Sigma-Aldrich | Cat #: P5318 | |
| Chemical compound, drug | Benzamidine hydrochloride hydrate | Sigma-Aldrich | Cat #: B6506 | |
| Chemical compound, drug | Phenylmethylsulfonyl fluoride (PMSF) | Sigma-Aldrich | Cat #: 10837091001 | |
| Chemical compound, drug | Aprotinin | Sigma-Aldrich | Cat #: A1153 | |
| Chemical compound, drug | Trypsin inhibitor | Sigma-Aldrich | Cat #: T9128 | |
| Chemical compound, drug | Bestatin | GoldBio | Cat #: B-915–100 | |
| Chemical compound, drug | E-64 | GoldBio | Cat #: E-064–25 | |
| Chemical compound, drug | Phosphoramidon disodium salt | Sigma-Aldrich | Cat #: R7385 | |
| Commercial assay, kit | 3–12% Bis-Tris Protein Gels | ThermoFisher Scientific | BN1003BOX | |
| Commercial assay, kit | NativePAGE Running Buffer Kit | ThermoFisher Scientific | BN2007 | |
| Commercial assay, kit | NativePAGE 5% G-250 Sample Additive | ThermoFisher Scientific | BN2004 | |
| Commercial assay, kit | NativePAGE Sample Buffer (4X) | ThermoFisher Scientific | BN2003 | |
| Software, algorithm | Slidebook | Intelligent imaging | RRID: SCR_014300 | |
| Software, algorithm | Fiji/ImageJ | Fiji | SCR_002285 | |
| Software, algorithm | FCS analysis tool | Smith Lab, University of Akron | | |

## Expression and purification

Genes encoding l- (residues 88–960) or s- (residues 195–960) OPA1 (UniProt O60313-1) were codon optimized for expression in *Pichia pastoris* and synthesized by GenScript (NJ, USA). The sequences encode Twin-Strep-tag, HRV 3C site, $(G_4S)_3$ linker at the N-terminus and $(G_4S)_3$ linker, TEV site, deca-histidine tag at the C-terminus. The plasmids were transformed into the methanol inducible SMD1163 strain (gift from Dr. Tom Rapoport, Harvard Medical School) and the clones exhibiting high Opa1 expression were determined using established protocols. For purification, cells expressing l-Opa1 were resuspended in buffer A (50 mM sodium phosphate, 300 mM NaCl, 1 mM 2-mercaptoethanol, pH 7.5) supplemented with benzonase nuclease and protease inhibitors and lysed using an Avestin EmulsiFlex-C50 high-pressure homogenizer. The membrane fractions were collected by ultracentrifugation at 235,000 x g for 45 min. at 4°C. The pellet was resuspended in buffer A containing 2% DDM, (Anatrace, OH, USA) 0.1 mg/ml 18:1 cardiolipin (Avanti Polar Lipids, AL, USA) and protease inhibitors and stirred at 4°C for 1 hr. The suspension was subjected to ultracentrifugation at 100,000 x g for 1 hr at 4°C. The extract containing l-Opa1 was loaded onto a Ni-NTA column (Biorad, CA, USA), washed with 40 column volumes of buffer B (50 mM sodium phosphate, 350 mM NaCl, 1 mM 2-mercaptoethanol, 1 mM DDM, 0.025 mg/ml 18:1 cardiolipin, pH 7.5) containing 25 mM imidazole and 60 column volumes of buffer B containing 100 mM imidazole. The bound protein was eluted with buffer B containing 500 mM imidazole, buffer exchanged into buffer C (100 mM Tris-HCl, 150 mM NaCl, 1 mM EDTA, 1 mM 2-mercaptoethanol, 0.15 mM DDM, 0.025 mg/ml 18:1 cardiolipin, pH 8.0). In all the functional assays, the C-terminal His tag was cleaved by treatment with TEV protease and passed over the Ni-NTA and Strep-Tactin XT Superflow (IBA Life Sciences, Göttingen, Germany) columns attached in tandem. The Strep-Tactin XT column was detached, washed with buffer C and eluted with buffer C containing 50 mM biotin. The elution fractions were

concentrated and subjected to size exclusion chromatography in buffer D (25 mM BIS-TRIS propane, 100 mM NaCl, 1 mM TCEP, 0.025 mg/ml 18:1 cardiolipin, pH 7.5, 0.01% LMNG, 0.001% CHS). s-OPA1 was purified using a similar approach but with one difference: post lysis, the DDM was added to the unclarified lysate at 0.5% concentration and stirred for 30 min. – 1 hr. at 4°C prior to ultracentrifugation. The supernatant was applied directly to the Ni-NTA column.

## GTPase activity assay

The GTPase activity of purified Opa1 was analyzed using EnzCheck Phosphate Assay Kit (Thermo Fisher, USA) according to the vendor's protocol. Each condition was performed in triplicate. The GTPase assay buffers contained 25 mM HEPES, 60 mM NaCl, 100 mM KCl, 0.5 mM $MgCl_2$ with 0.15 mM DDM. 60 µM GTP was added immediately before data collection. To compare the effect of cardiolipin on GTPase activity, additional 0.5 mg/ml Cardiolipin was dissolved in the reaction buffer and added to the reaction to a final concentration of 0.02 mg/ml. The absorbance at 340 nm of each reaction mixture was recorded using SpectraMax i3 plate reader (Molecular Devices) every 30 s. Experiments were performed in triplicate. Resulting Pi concentration was fitted to a single-phase exponential-decay, specific activity data were fitted to a Michaelis-Menten equation (GraphPad Prism 8.1).

## Preparation of polymer-tethered lipid bilayers

Lipid reagents, including 1,2-dioleoyl-sn-glycero-3-phosphocholine, (DOPC); 1,2-dioleoyl-sn-glycero-3-phosphoethanolamine-N-[methoxy(polyethylene glycol)−2000] (DOPE-PEG2000), L-α-phosphatidylinositol (Liver PI) and 1′,3′-bis[1,2-dioleoyl-sn-glycero-3-phospho]-glycerol (cardiolipin) were purchased from Avanti Polar Lipids (AL, USA). To fabricate the polymer-tethered lipid bilayers, we combined Langmuir-Blodgett and Langmuir-Schaefer techniques, using a Langmuir-Blodgett Trough (KSV-NIMA, NY, USA) (*Siegel et al., 2011*; *Ge et al., 2014*). For cardiolipin-free lipid bilayers, a lipid mixture with DOPC with 5% (mol) DOPE-PEG2000% and 0.2% (mol) Cy5-DSPE at the total concentration of 1 mg/ml was spread on the air water interface in a Langmuir trough. The surface pressure was kept at 30 mN/m for 30 min before dipping. The first lipid monolayer was transferred to the glass substrate (25 mm diameter glass cover slide, Fisher Scientific, USA) through Langmuir-Blodgett dipping, where the dipper was moved up at a speed of 22.5 mm/min. The second leaflet of the bilayer was assembled through Langmuir-Schaefer transfer after 1 mg/ml of DOPC with 0.2% (mol) Cy5-PE (Avanti Polar Lipids, AL, USA) was applied to an air-water interface and kept at a surface pressure of 30 mN/m.

Lipid bilayer with cardiolipin was fabricated in a similar manner, where the bottom leaflet included 7% (mol) Liver PI, 20% (mol) cardiolipin, 20% (mol) DOPE, 5% (mol) DOPE-PEG2000, 0.2% (mol) Cy5-PE and 47.8% DOPC. The composition of the top leaflet of the bilayer was identical except for the absence of DOPE-PEG2000. To match the area/molecule at the air-water interface between CL-free and CL-containing bilayer, the film pressure was kept at 37 mN/m. The final average area per lipid, which is the key factor affecting lipid lateral mobility, was kept constant at a $A_{lipid} = 65$ Å$^2$ (*Lewis and McElhaney, 2009*).

Double bilayers were fabricated according to previous reports (*Minner et al., 2013*). The first bilayer containing DOPC with 5% (mol) DSPE-PEG2000-Maleimide (Avanti Polar Lipids, AL, USA) and 0.2% (mol) Cy5-DOPE in both inner and outer leaflets was made using Langmuir-Blodgett/Langmuir-Schaefer methods. The second planar lipid bilayer was formed by fusion of lipid vesicles and removal of non-fused vesicles. Lipid vesicles were formed by hydrating dried lipid films with DOPC, 0.2% (mol) TexasRed-DHPE and 5% (mol) of linker lipid (DPTE, AL, USA) in a 0.1 mM sucrose/1 mM $CaCl_2$ solution. The lipid suspension was heated for 1.5 hr at 75°C, and added to the first bilayer in a 0.1 mM glucose/1 mM $CaCl_2$ solution. After 2 hr of incubation, additional vesicles were removed by extensive rinsing. The bilayer was then imaged by TIRF microscope.

## Reconstitution of l-Opa1 into lipid bilayers

Purified l-Opa1 was first desalted into 25 mM Bis-Tris buffer with 150 mM NaCl containing 1.2 nM DDM and 0.4 µg/L of cardiolipin to remove extra surfactant during purification. The resulting protein was added to each bilayer to the total amount of $1.3 \times 10^{-12}$ mol (protein:lipid 1:10000) together with a surfactant mixture of 1.2 nM of DDM and 1.1 nM n-Octyl-β-D-Glucopyranoside (OG,

Anatrace, OH, USA). The protein was incubated for 2 hr before removal of the surfactant. To remove the surfactant, Bio-Beads SM2 (Bio-Rad, CA, USA) was added to the solution at a final concentration of 10 μg beads per mL of solution and incubated for 10 min. Buffer with 25 mM Bis-Tris and 150 mM NaCl was applied to remove the Bio-beads with extensive washing. Successful reconstitution was determined using fluorescent correlation spectroscopy assay as described in the supplemental materials.

## Preparation of liposomes and proteoliposomes

To prepare calcein (MilliporeSigma, MA, USA) encapsulated liposomes, lipid mixtures (7% (mol) PI, 20% cardiolipin, 20% PE, 0.2% TexasRed-PE, DOPC (52.8%)), were dissolved in chloroform and dried under argon flow for 25 min. The resulting lipid membrane was mixed in 25 mM Bis-Tris with 150 mM NaCl and 50 mM calcein through vigorous vortexing. Lipid membranes were further hydrated by incubating the mixtures under 70°C for 30 min. Large unilamellar vesicles (LUVs) were prepared by extrusion (15 to 20 times) using a mini-extruder with 200 nm polycarbonate membrane.

Proteoliposomes were prepared by adding purified l-Opa1 in 0.1 μM DDM to prepared liposomes at a protein: lipid of 1:5000 (2.5 μg l-Opa1 for 0.2 mg liposome) and incubated for 2 hr. Surfactant was removed by dialysis overnight under 4°C using a 3.5 KDa dialysis cassette. Excess calcein was removed using a PD-10 desalting column. The final concentration of liposome was determined by TexasRed absorbance, measured in a SpectraMax i3 plate reader (Molecular Devices).

To evaluate l-Opa1 reconstitution into proteoliposomes, dye free liposome was prepared with TexasRed conjugated anti-His tag Antibody (ThermoFisher) by mixing lipids with antibody containing buffer. TexasRed Labeling efficiency of the antibody was calculated to be 1.05 according to the vendor's protocol. Antibodies were added at a concentration of 2.6 μg/ml to 0.2 mg/ml liposome. Following hydration through vortexing at room temperature for 15 min, Large unilamellar vesicles were formed following 20 times extrusion procedure described above. Liposomes labeled with 0.02% (mol) TexasRed-PE were also prepared as a standard for quantifying reconstitution rate.

For the co-flotation analysis, 200 μl of 20 mg/ml TexasRed-DHPE (0.2% (mol)) labeled proteoliposome (reconstitution ratio, protein:lipid 1:5000) was loaded to sucrose gradient (with steps of 0%, 15%, 30%, 60%). The volume of each fraction was 800 μl. Sucrose solutions were prepared in Bis-Tris buffer (25 mM Bis-Tris, 150 mM NaCl, pH 7.4). Samples were then centrifuged using a high-speed centrifuge equipped with SW 55i rotor (Beckmann Coulter, CA, USA) for 2.5 hr at a speed of 30000 xg. For high salt and carbonate treatment, the same amount of proteoliposome was redistributed in Bis-Tris buffer with 500 mM NaCl (pH 7.4) and buffer containing 50 mM $Na_2CO_3$ and 50 mM NaCl (pH 8.2), respectively. The resulting suspension was loaded in gradient for separation. After centrifugation, all fractions were collected and concentrated to 40 μl. Fractions were detected by western blot and then analyzed by ImageJ. The presence of liposomes was detected by absorbance at 590 nm using a DeNovix FX photometer (DeNovix, Inc).

## Fluorescent correlation spectroscopy

Fluorescence correlation spectroscopy (FCS) was performed using a home-built PIE-FCCS system (*Huang et al., 2016*; *Comar et al., 2014*). Two pulsed laser beams with wavelengths of 488 nm (9.7 MHz, five ps) and 561 nm (9.7 MHz, five ps) were filtered out from a supercontinuum white light fiber laser (SuperK NKT Photonics, Birkerod, Denmark) and used as excitation beams. The laser beams were sent through a 100X TIRF objective (NA 1.47, oil, Nikon Corp., Tokyo, Japan) to excite the samples in solution or on bilayer. The emission photons were guided through a common 50 μm diameter pinhole. The light was spectrally separated by a 560 nm high-pass filter (AC254-100-A-ML, Thorlabs), further filtered by respective bandpass filters (green, 520/44 nm [FF01-520/44-25]; red, 612/69 nm [FF01-621/69-25], Semrock), and finally reach two single photon avalanche diode (SPAD) detectors (Micro Photon Devices). The synchronized photon data were collected using a time correlated single photon counting (TCSPC) module (PicoHarp 300, PicoQuant, Berlin, Germany).

The collected photon data were transformed into correlation functions with a home written MATLAB code. The correlation functions were fitted using two-dimensional (*Hoppins et al., 2007*) or three-dimensional (*Cipolat et al., 2006*) Brownian diffusion model for bilayer or solution samples respectively.

$$G(\tau) = \frac{1}{\langle N \rangle} \frac{1}{1 + \frac{\tau}{\tau_D}} \qquad (1)$$

$$G(\tau) = \frac{1}{\langle N \rangle} \frac{1}{1 + \frac{\tau}{\tau_D}} \frac{1}{\sqrt{1 + \omega^2 \cdot \frac{\tau}{\tau_D}}} \qquad (2)$$

Where N is the average number of particles in the system, $\omega$ is the waist of the excitation beam, and $\tau_D$ is the dwell time that can be used to calculate the diffusion coefficient (D) of the particles (*Huang et al., 2016*).

$$\tau_D = \frac{\omega^2}{4D}$$

Measurements were made on buffers with evenly distributed liposomes, proteoliposomes and antibodies in a glass-bottom 96 well plate at room temperature. The plates were pre-coated with lipid bilayer fabricated from 100 nm DOPC liposomes. For each solution, data were collected in five successive 15 s increments.

For characterization of l-Opa1 reconstitution into planar bilayers, an anti-Opa1 C-terminal antibody (Novus Biologicals, CO, USA) was used. The antibody was labeled by TexasRed (Texas Red-X Protein Labeling Kit, ThermoFisher, CA, USA). Labeling efficiency of the antibody was determined as 1.52 TexasRed/antibody, as determined by NanoDrop (ThermoFisher, CA, USA). The labeled antibody was added to l-Opa1 in the supported bilayer at twice the total introduced Opa1 concentration. Excess antibody was removed by extensive rinsing.

To estimate reconstitution efficiency, 0.002% (mol) l-Opa1 was added to the bilayer. In a separate experiment 0.002% (mol) TexasRed-PE was introduced to the bilayer. The reconstitution efficiency was calculated from the anti-l-Opa1 antibody TexasRed fluorophore density divided by the TexasRed-PE fluorphore density, normalized by the antibody labeling efficiency (1.5 dye molecules/antibody).

## Total Internal Reflection Fluorescent Microscopy (TIRF)

Liposome docking and lipid exchange events were imaged using a Vector TIRF system (Intelligent Imaging Innovations, Inc, Denver, CO, USA) equipped with a W-view Gemini system (Hamamatsu photonics, Bridgewater, NJ). TIRF images were acquired using a 100X oil immersion objective (Ziess, N.A 1.4). A 543 nm laser was used for the analysis of TexasRed-PE embedded liposomes and proteoliposomes, while a 633 nm laser was applied for the analysis of Cy5-PE embedded in the planar lipid bilayer. Fluorescent emission was simultaneously observed through a 609-emission filter with a band width of 40 nm and a 698-emission filter with a band width of 70 nm. The microscope system was equipped with a Prime 95B scientific CMOS camera (Photometrics), maintained at −10°C. Images were taken at room temperature, before adding any liposome or proteoliposome, after 15 mins of addition, and after 30 mins of adding GTP (1 mM) and MgCl2 (1 mM). Each data point was acquired from five different bilayers, each bilayer data contains 5–10 particles on average.

Dwell times for hemifused particles were recorded from the moment of GTP addition for pretethered particles, until the time of half-maximal TexasRed signal decay. Full fusion events were recorded by monitoring the calcein channel at particle locations identified through the TexasRed signal. Particle identification and localization used both uTrack (*Jaqaman et al., 2008*) and Slidebook (Intelligent Imaging Innovations, Inc, Denver, CO) built-in algorithms. To calibrate the point spread function 100 nm and 50 nm fluorescent particles (ThermoFisher Scientific) were used. 2D Gaussian detection was applied in both cases. 2-way ANOVA tests were done using GraphPad Prism. Intensity and distribution of the particles were analyzed using ImageJ.

For analysis of protein reconstitution in proteoliposome (stoichiometry), a TIRF microscope modified from an inverted microscope (Nikon Eclipse Ti, Nikon Instruments) was used. A 561 nm diode laser (OBIS, Coherent Inc, Santa Clara, USA) was applied at TIRF angle through a 100X TIRF objective (NA 1.47, oil, Nikon) and the fluorescence signals were collected by an EMCCD camera (Evolve 512, Photometrics).

## Nanosight NTA analysis

A NTA300 Nanosight instrument was used to evaluate size distribution of liposome and proteoliposome under different conditions. The equipment was equipped with a 405 nm laser and a CMOS camera. 1 ml of 0.1 µg/ml sample was measured, to reach the recommended particle number of $1 \times 10^8$ particles/mL (corresponding to the dilution factor of 1:100,000). Image acquisition was conducted for 40 s for each acquisition and repeated for 10 times for every injection. Three parallel samples were examined for the determination of size distribution. Under each run, the camera level was set to 12 and the detection threshold was set at 3.

## Blue native polyacrylamide gel electrophoresis (BN-PAGE)

Bis-Tris gradient gels (3–12%) were purchased from ThermoFisher Scientific (Cat. No. BN1003BOX) and BN-PAGE was performed according to manufacturer's instructions. Gel samples (10 µl) were prepared by mixing indicated quantity of Opa1 with sample buffer containing 0.25% Coomassie G-250 and 1 mM DDM. For experiments involving l-Opa1 and s-Opa1 mixtures, the samples were incubated on ice for 10 min before loading. The cathode buffer contained 1 mM DDM and electrophoresis was performed at 4°C with an ice jacket surrounding the apparatus.

## Acknowledgements

We thank members of the Chao lab for helpful discussions and review of the manuscript. LHC is grateful for support from a Charles H Hood Foundation Child Health Research Award. We thank Fanny Ng and the Szostak Lab for technical support. Work by XS and AWS are supported by the National Science Foundation under Grant No. CHE-1753060.

## Additional information

### Funding

| Funder | Grant reference number | Author |
| --- | --- | --- |
| Charles H. Hood Foundation | Child Health Research Award | Yifan Ge Luke H Chao |
| National Science Foundation | CHE-1753060 | Xiaojun Shi Adam W Smith |

The funders had no role in study design, data collection and interpretation, or the decision to submit the work for publication.

### Author contributions

Yifan Ge, Conceptualization, Resources, Data curation, Formal analysis, Validation, Investigation, Methodology; Xiaojun Shi, Resources, Data curation, Software, Formal analysis, Validation, Investigation, Methodology; Sivakumar Boopathy, Julie McDonald, Resources, Validation, Investigation, Methodology; Adam W Smith, Conceptualization, Resources, Software, Supervision, Funding acquisition, Methodology, Project administration; Luke H Chao, Conceptualization, Formal analysis, Supervision, Funding acquisition, Investigation, Project administration

### Author ORCIDs

Yifan Ge (iD) https://orcid.org/0000-0001-9135-9569
Xiaojun Shi (iD) https://orcid.org/0000-0002-8060-5880
Sivakumar Boopathy (iD) https://orcid.org/0000-0003-0524-3338
Julie McDonald (iD) http://orcid.org/0000-0002-3715-9619
Adam W Smith (iD) http://orcid.org/0000-0001-5216-9017
Luke H Chao (iD) https://orcid.org/0000-0002-4849-4148

### Decision letter and Author response
Decision letter https://doi.org/10.7554/eLife.50973.sa1

Author response https://doi.org/10.7554/eLife.50973.sa2

## Additional files

### Supplementary files
- Transparent reporting form

### Data availability
All data generated or analyses during this study are include in the manuscript and supporting files.

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
