## [Decision Letter]

**Acceptance summary:**

The work presented addresses the molecular mechanism of inner membrane fusion mediated by the mammalian dynamin related GTPase, Opa1. The experiments use a reconstituted in vitro fusion system that distinguishes between the key steps in membrane fusion: tethering, membrane docking, lipid mixing (hemifusion), and content mixing. Among the key findings are that both l-Opa1 and s-Opa1 are required for efficient fusion.

**Decision letter after peer review:**

Thank you for submitting your article "Two forms of Opa1 cooperate to complete fusion of the mitochondrial inner-membrane" for consideration by *eLife*. Your article has been reviewed by three peer reviewers, and the evaluation has been overseen by Axel Brunger as the Reviewing Editor and Suzanne Pfeffer as the Senior Editor. The reviewers have opted to remain anonymous.

The reviewers have discussed the reviews with one another and the Reviewing Editor has drafted this decision to help you prepare a revised submission.

Summary:

The work presented addresses the molecular mechanism of inner membrane fusion mediated by the mammalian dynamin related GTPase, Opa1. The experiments use a reconstituted in vitro fusion system that distinguishes between the key steps in membrane fusion: tethering, membrane docking, lipid mixing (hemifusion), and content mixing. Among the key findings are that both l-Opa1 and s-Opa1 are required for efficient fusion. While the findings are of considerable interest, some additional control experiments interrogating reconstituted l- and s-Opa1 isoforms are required. In addition, the results should be better related to the other results in the literature.

Major revisions:

Control experiments assessing the fidelity of reconstitution are required, for example, is l-Opa1 integral or peripherally associated with membranes? Is it salt/carbonate resistant? This is critical given peripheral l-Opa1 likely will function like s-Opa1 both by its intrinsic GTPase activities/assembly properties and by exchanging between membranes. This control would strengthen the validity of all of the conclusions derived from l-Opa1 reconstitution experiments: homotypic tethering, heterotypic-CL tethering, rations of L-Opa1/s-Opa1, etc.

Previous work (both in vitro and in vitro) suggested that membrane integrated l-Mgm1 possesses no GTPase activity and that GTPase activity is not required for fusion, respectively. These previous results are now contradicted by the data presented in Figure 2. As control, please address the role of GTPase activity in s-Opa1 and l-Opa1 for tethering and fusion using mutant proteins that interfere with GTPase activity.

Please comment why in Figure 3 tethering is observed even in the absence of GTP and Cardiolipin, which are both described to be necessary for Opa1 membrane fusion. In Figure 3B when incubated with nonhydrolyzable GTP analog one can see much lower tethering than in basal (apo) state. Is Opa1 able to tether liposomes by its own?

Based on Figure 2—figure supplement 3, the authors claim that Opa1 has the potential to self-associate and oligomerize in their reconstituted system. Please perform blue native page (BN PAGE) to verify oligomerization and association.

s-Opa1 tethering in Figure 3D and no FRET in the same situation in Figure 4 is puzzling. If s-Opa1 causes tethering and membranes must be within 40Å for FRET, then the complex of s-Opa1 must be bigger or there is already hemifusion as shown in Figure 5, which is causing the FRET to work. Is there any possible "proof of principle" experiment, which would show FRET in tethered liposomes without hemifusion? Please perform BN PAGE of l-Opa1 and s-Opa1 to verify if s-Opa1 is not inducing some bigger protein complexes. Electron microscopy to show different tethering of l-Opa1 and s-Opa1 would be desirable, but not essential.

In the last result section, high concentrations of s-Opa1 inhibit fusion by disruption of the l-Opa1:l-Opa1 interaction. Please perform BN-PAGE with increasing concentration of s-Opa1 to see if s-Opa1 really binds to the complex l-Opa dimers.

Textual revisions:

The Introduction and Discussion are inadequate in terms of presenting an overview of the existing knowledge regarding Opa1 mechanism and function. At a minimum, the Introduction should include recent structural work on Mgm1 published by the Daumke group and present functional work from the Langer and Chan labs on the roles of long and short Opa1 isoforms. Indeed, the primary contribution of the work is providing definitive evidence that both Opa1 isoforms are required for fusion, which directly refutes observations in cells from the Langer lab that l-Opa1 is sufficient for fusion. This point should be discussed in detail. There is much debate in the field as to the requirements of s and l forms of Opa1 in fusion. The Chan lab also has data indicating that Opa1 processing is coupled to and required for fusion. In addition, work in yeast showed that the long isoform stimulated the GTPase activity of the short isoform. The justification statement in the Introduction that the "…activities of the two forms and their regulator interplay remain unclear" is not accurate.

[Editors' note: further revisions were suggested prior to acceptance, as described below.]

Thank you for resubmitting your work entitled "Two forms of Opa1 cooperate to complete fusion of the mitochondrial inner-membrane" for further consideration by *eLife*. Your revised article has been evaluated by Suzanne Pfeffer (Senior Editor) and a Reviewing Editor.

The manuscript has been improved but there are some remaining issues that need to be addressed In the final revision before acceptance, as outlined below.

Reviewer #1:

The manuscript is greatly improved. Many of the technical concerns have now been addressed. In terms of assessing GTPase requirements of l- and s-Opa1, did the authors test G300E l-Opa1 with wt s-Opa1 for membrane tethering and fusion activity? This requested experiment is not apparently presented and was one of the key experiments requested i.e. what is the role of the l-GTPase domain?

The addition of relevant previous findings from the Langer and Chan groups also improved the manuscript. However, the language used regarding the key finding of the Chan group – that OPA1 processing per se is required for fusion – is not accurately described. The authors must clarify between the requirement for s-OPA1 versus l-OPA1 processing to s-OPA1 for fusion. Please see below for the passages/edits that need to be revised for accuracy on this point.

Perhaps the origin for this confusion comes from the following statement:

“Since Yme1L activity is tied to respiratory state, supplying cells with substrates for oxidative phosphorylation shifts the mitochondrial network to a more tubular state. These observations led the Chan group to conclude that Opa1 processing is important for fusion.”

This statement is not entirely correct. In vitro analysis of mitochondrial fusion using protease inhibitors etc indicated that processing per se was required for fusion.

“These directly conflicting interpretations of cellular observations have remained unreconciled. Is proteolytic processing of Opa1 required for regulating fusion, and if so, is the processing stimulatory or inhibitory?”

Suggested revision: Is proteolytic processing of Opa1 required for regulating fusion? Is s-Opa1 required for fusion?

“Our model proposes that l-Opa1:s-Opa1 stoichiometry, resulting from proteolytic processing, gates the final step of fusion, pore opening.”

Suggested revision: Our model proposes that l-Opa1:s-Opa1 stoichiometry gates the final step of fusion, pore opening.

“In contrast to the Langer group's conclusions, we find that Opa1 processing strongly stimulates fusion activity, as observed by the Chan and colleagues.”

Suggested revision: In contrast to Chan's conclusions, we find that s-Opa1 strongly stimulates l-Opa1 dependent fusion activity independent of the Yme1 processing reaction.

---

## [Author Response]

Major revisions:Control experiments assessing the fidelity of reconstitution are required, for example, is l-Opa1 integral or peripherally associated with membranes? Is it salt/carbonate resistant? This is critical given peripheral l-Opa1 likely will function like s-Opa1 both by its intrinsic GTPase activities/assembly properties and by exchanging between membranes. This control would strengthen the validity of all of the conclusions derived from l-Opa1 reconstitution experiments: homotypic tethering, heterotypic-CL tethering, rations of L-Opa1/s-Opa1, etc.

We thank the reviewers for suggesting this important control, and now include data showing that l-Opa1 reconstituted into cardiolipin-containing liposomes is indeed salt and carbonate-resistant (Figure 2—figure supplement 2A-C). We show that l-Opa1 reconstituted into cardiolipin-free (DOPC) liposomes show similar salt and carbonate-resistant behavior (Figure 2—figure supplement 2D-F), indicating reconstitution is not dependent on cardiolipin in the liposomes. Finally, we show s-Opa1 membrane liposome association is dependent on the presence of cardiolipin, as s-Opa1 co-floats with cardiolipin liposomes (Figure 2—figure supplement 2G), but does not co-float with DOPC liposomes (Figure 2—figure supplement 2J). High salt treatment did not show a dramatic effect on the s-Opa1 cadiolipin interaction (Figure 2—figure supplement 2H), consistent with a hydrophobic, rather than electrostatic, interaction by the membrane ‘paddle’-mediated association. However, carbonate treatment disrupts s-Opa1 association with cardiolipin liposomes (Figure 2—figure supplement 2I). These control observations are consistent with successful reconstitution of l-Opa1 into liposomes, and peripheral membrane association of s-Opa1, which together support the use of l-Opa1 proteoliposomes for this study.

Previous work (both in vitro and in vitro) suggested that membrane integrated l-Mgm1 possesses no GTPase activity and that GTPase activity is not required for fusion, respectively. These previous results are now contradicted by the data presented in Figure 2. As control, please address the role of GTPase activity in s-Opa1 and l-Opa1 for tethering and fusion using mutant proteins that interfere with GTPase activity.

The material used in Figure 2 was solubilized in DDM during purification, and the GTPase activity reported is for detergent-solubilized material. Detergent-solubilized WT l-Opa1 material shows activity dependent on the presence of cardiolipin (Figure 2C and D). We have now added additional experimental data showing the GTPase activity for WT s-Opa1, G300E l-Opa1, and G300E s-Opa1 (Figure 2—figure supplement 1A-D). We note that detergent-solubilized material is free from membrane constraints, and these regulatory interactions due to association with the membrane may be lost. Attempts to measure GTPase activity of our liposome-reconstituted material in the EnzCheck Phosphate Assay assays were unsuccessful due to background from the liposomes.

We performed an additional set of experiments testing the role of GTPase activity on tethering. WT l-Opa1 tethering is disrupted by the presence of G300E s-Opa1 (Figure 3—figure supplement 1A). Increasing amounts of G300E s-Opa1 results in l-Opa1 tethered liposomes detaching from the l-Opa1-containing bilayer. As previously observed, the hydrolysis-dead mutant G300E l-Opa1 alone does not tether liposomes to a supported bilayer (Figure 3—figure supplement 1B). Also, G300E l-Opa1 in the presence of equimolar amounts of G300E s-Opa1 does not induce membrane tethering (Figure 3—figure supplement 1B).

We also investigated the potential for l-Opa1 to potentiate the activity of s-Opa1. Using detergent solubilized material, we did not see dramatic enhancement of s-Opa1 activity in the presence of G300E l-Opa1 (Figure 2—figure supplement 1F). There is also little effect for WT l-Opa1 activity in the presence of G300E s-Opa1 (Figure 2—figure supplement 1E). It is worth noting that we use a different GTPase-dead mutant from the S224A mutant used in previous studies (DeVay et al., 2009) that showed l-Opa1 potentiation of s-Opa1.

Please comment why in Figure 3 tethering is observed even in the absence of GTP and Cardiolipin, which are both described to be necessary for Opa1 membrane fusion. In Figure 3B when incubated with nonhydrolyzable GTP analog one can see much lower tethering than in basal (apo) state. Is Opa1 able to tether liposomes by its own?

Our experiments implicate interactions outside of the GTP-ase domains for tethering. These interactions may include one of the stalk interfaces observed by Faelber et al. The accessibility of these interfaces, may be dependent on nucleotide state. Since tethering is an obligate step for fusion, under these conditions, we do not observe any hemifusion or pore opening events. We find that l-Opa1 does not tether naked liposomes lacking l-Opa1 or cardiolipin, indicating l-Opa1 alone is unable to tether liposomes. This observation also indicates the tethering effects we observe are not due to defects induced by reconstitution that could cause liposomes to stick to an exposed glass regions (Figure 4—figure supplement 1B).

Based on Figure 2—figure supplement 3, the authors claim that Opa1 has the potential to self-associate and oligomerize in their reconstituted system. Please perform blue native page (BN PAGE) to verify oligomerization and association.

We now include BN-PAGE of the proteins used in this study (Figure 2—figure supplement 4A). Both l-Opa1 and s-Opa1 run as an oligomeric mixture, with a major species slightly above and below ~480KDa, respectively. For this detergent solubilized material, the orientation relative to the bilayer may be an important factor influencing self-assembly. These observations support the claim that the proteins have the potential to self-associate. We observe that these proteins freely diffuse (at much lower concentrations) in the membranes, as observed by FCS.

s-Opa1 tethering in Figure 3D and no FRET in the same situation in Figure 4 is puzzling. If s-Opa1 causes tethering and membranes must be within 40Å for FRET, then the complex of s-Opa1 must be bigger or there is already hemifusion as shown in Figure 5, which is causing the FRET to work. Is there any possible "proof of principle" experiment, which would show FRET in tethered liposomes without hemifusion?

We have now included a “proof of principle" experiment comparing FRET signals for intra-bilayer FRET and inter-bilayer FRET (signal between two supported lipid bilayers tethered by PEG), in the absence of hemifusion (Figure 4—figure supplement 1A). Under our imaging conditions and microscope settings, we observe low FRET for membranes tethered with a distance of ~7 nm, in contrast with high levels of intra-bilayer FRET. Both homotypic and heterotypic arrangements of l-Opa1 are capable of inducing moderate (~40%) levels of FRET between bilayers. Bilayers tethered by s-Opa1 alone show low levels of FRET. Recent crystal structures of s-Mgm1 from Faelber et al. show the distances of the paddle domains to be ~120 Å. A dimer bridging two bilayers in this arrangement would be expected to have low FRET signal.

Please perform BN PAGE of l-Opa1 and s-Opa1 to verify if s-Opa1 is not inducing some bigger protein complexes. Electron microscopy to show different tethering of l-Opa1 and s-Opa1 would be desirable, but not essential.

As noted, BN-PAGE of s-Opa1 and l-Opa1 show major species slightly above and below ~480KDa, respectively (Figure 2—figure supplement 4A). The size of this assembly (consistent with a tetramer), may allow for tethering interactions in the absence of FRET (discussed above). Electron microscopy analysis of l-Opa1 and s-Opa1 tethering complexes on liposomes is an important next step, but outside the scope of this study.

In the last result section, high concentrations of s-Opa1 inhibit fusion by disruption of the l-Opa1:l-Opa1 interaction. Please perform BN-PAGE with increasing concentration of s-Opa1 to see if s-Opa1 really binds to the complex l-Opa dimers.

We note that a strength of the supported bilayer system is the ability to orient molecules in a manner expected at a site of membrane-membrane contact. Our competition experiments show that the tethered state can be competed for with G300E s-Opa1 (Figure 3—figure supplement 1A). We also performed the BN-PAGE titration (increasing the concentration of s-Opa1, in the presence of l-Opa1 as suggested), but found results inconclusive (Figure 2—figure supplement 4B). Addition of s-Opa1 to l-Opa1 results in a mixture of species from ~480 KDa to 1 MDa. This pattern is also seen with G300E l-Opa1 + WT s-Opa1. We are not able to distinguish the composition of the higher order species (whether there is both s-Opa1 and l-Opa1), because they were purified with the same tags, but note that in the presence of excess G300E s-Opa1, there is a slight enrichment of a ~700 KDa species in the presence of l-Opa1. The range of oligomeric species seen in the BN-PAGE gels is consistent with larger oligomers forming at high protein concentrations. Since BN-PAGE of detergent solubilized material will release the constraints of the membrane, we cannot specifically distinguish whether a tethered state is disrupted using this method.

Textual revisions:The Introduction and Discussion are inadequate in terms of presenting an overview of the existing knowledge regarding Opa1 mechanism and function. At a minimum, the Introduction should include recent structural work on Mgm1 published by the Daumke group and present functional work from the Langer and Chan labs on the roles of long and short Opa1 isoforms. Indeed, the primary contribution of the work is providing definitive evidence that both Opa1 isoforms are required for fusion, which directly refutes observations in cells from the Langer lab that l-Opa1 is sufficient for fusion. This point should be discussed in detail. There is much debate in the field as to the requirements of s and l forms of Opa1 in fusion. The Chan lab also has data indicating that Opa1 processing is coupled to and required for fusion. In addition, work in yeast showed that the long isoform stimulated the GTPase activity of the short isoform. The justification statement in the Introduction that the "…activities of the two forms and their regulator interplay remain unclear" is not accurate.

We have removed the justification statement and revised the Introduction to highlight important recent structural work from the Daumke group. The previous functional work from the Chan and Langer labs is now discussed in detail in both the Introduction and Discussion, as to relate it with this study. We thank the reviewers for the chance to better place this study in context with previous work.

[Editors' note: further revisions were suggested prior to acceptance, as described below.]

Reviewer #1:The manuscript is greatly improved. Many of the technical concerns have now been addressed. In terms of assessing GTPase requirements of l- and s-Opa1, did the authors test G300E l-Opa1 with wt s-Opa1 for membrane tethering and fusion activity? This requested experiment is not apparently presented and was one of the key experiments requested ie what is the role of the l-GTPase domain?

We assessed the GTPase activity for G300E l-Opa1 with WT s-Opa1 in our previous revision. In Figure 2—figure supplement 1 we measured the GTPase activity of G300E l-Opa1 with WT s-Opa1 and found the specific activity to be similar to WT-s-Opa1 alone, suggesting the l-Opa1 GTPase domain is not stimulating activity of the s-Opa1 GTPase domain. As we noted in our previous response, this mixture of l-Opa1 and s-Opa1 is different from previous comparisons (DeVay et al., 2009), because of the different mutant used (S224A), and the fact that we measured the activity of detergent solubilized material.

In this new revision, we include data for G300E l-Opa1 membrane tethering in the presence of WT s-Opa1, in a new Figure 3—figure supplement 1, panel C. We apologize for the omission. We find that G300E l-Opa1 alone does not tether liposomes, as previously described Figure 3—figure supplement 1, panel B. As we add WT s-Opa1 to G300E l-Opa1 we do see an increased number of liposomes tethered to the supported bilayer. At 1:1 G300E l-Opa1:WT s-Opa1, we do not see any full fusion events. We note, however, that s-Opa1 alone tethers liposomes to a supported bilayer (Figure 3E), so we cannot exclude that these liposomes are interacting through s-Opa1 only, nor can distinguish from this experiment which liposomes are tethering via a G300E l-Opa1:WT s-Opa1 heterocomplex. In order to address these questions, we require independently labeled variants of the two forms, so that we can relate tethering and fusion with the composition of proteins present. This line of experiments is of great interest to us, but outside the scope of this study.

The addition of relevant previous findings from the Langer and Chan groups also improved the manuscript. However, the language used regarding the key finding of the Chan group – that OPA1 processing per se is required for fusion – is not accurately described. The authors must clarify between the requirement for s-OPA1 versus l-OPA1 processing to s-OPA1 for fusion. Please see below for the passages/edits that need to be revised for accuracy on this point.

We thank the reviewer for emphasizing this point and worked to ensure the manuscript clearly articulates this important past finding. We edited the language, in all passages indicated, to clarify and distinguish between the requirement for s-Opa1, versus l-Opa1 processing to s-Opa1 for fusion.

Perhaps the origin for this confusion comes from the following statement:“Since Yme1L activity is tied to respiratory state, supplying cells with substrates for oxidative phosphorylation shifts the mitochondrial network to a more tubular state. These observations led the Chan group to conclude that Opa1 processing is important for fusion.”This statement is not entirely correct. In vitro analysis of mitochondrial fusion using protease inhibitors etc indicated that processing per se was required for fusion.

This statement is now modified, and reads: Since Yme1L activity is tied to respiratory state, supplying cells with substrates for oxidative phosphorylation shifts the mitochondrial network to a more tubular state. Importantly, Chan and colleagues cleanly demonstrate, with an in vitro purified mitochondria system using protease inhibitors and an engineered cleavage site that mitochondrial fusion is dependent on proteolytic processing.

“These directly conflicting interpretations of cellular observations have remained unreconciled. Is proteolytic processing of Opa1 required for regulating fusion, and if so, is the processing stimulatory or inhibitory?”Suggested revision: Is proteolytic processing of Opa1 required for regulating fusion? Is s-Opa1 required for fusion?

Done.

“Our model proposes that l-Opa1:s-Opa1 stoichiometry, resulting from proteolytic processing, gates the final step of fusion, pore opening.”Suggested revision: Our model proposes that l-Opa1:s-Opa1 stoichiometry gates the final step of fusion, pore opening.

Done.

“In contrast to the Langer group's conclusions, we find that Opa1 processing strongly stimulates fusion activity, as observed by the Chan and colleagues.”Suggested revision: In contrast to Chan's conclusions, we find that s-Opa1 strongly stimulates l-Opa1 dependent fusion activity independent of the Yme1 processing reaction.

Done.